# Automated analysis of high-content microscopy data with deep learning

Oren Z Kraus[1,2,†] ⓘ, Ben T Grys[2,3,†], Jimmy Ba[1], Yolanda Chong[4], Brendan J Frey[1,2,5,6], Charles Boone[2,3,5,*] & Brenda J Andrews[2,3,5,**] ⓘ

## Abstract

Existing computational pipelines for quantitative analysis of high-content microscopy data rely on traditional machine learning approaches that fail to accurately classify more than a single dataset without substantial tuning and training, requiring extensive analysis. Here, we demonstrate that the application of deep learning to biological image data can overcome the pitfalls associated with conventional machine learning classifiers. Using a deep convolutional neural network (DeepLoc) to analyze yeast cell images, we show improved performance over traditional approaches in the automated classification of protein subcellular localization. We also demonstrate the ability of DeepLoc to classify highly divergent image sets, including images of pheromone-arrested cells with abnormal cellular morphology, as well as images generated in different genetic backgrounds and in different laboratories. We offer an open-source implementation that enables updating DeepLoc on new microscopy datasets. This study highlights deep learning as an important tool for the expedited analysis of high-content microscopy data.

**Keywords** deep learning; high-content screening; image analysis; machine learning; *Saccharomyces cerevisiae*
**Subject Categories** Computational Biology; Methods & Resources
**Mol Syst Biol. (2017) 13: 924**

## Introduction

Advances in automated image acquisition and analysis, coupled with the availability of reagents for genome-scale perturbation, have enabled systematic analyses of cellular and subcellular phenotypes (Mattiazzi Usaj *et al*, 2016). One powerful application of microscopy-based assays involves assessment of changes in the subcellular localization or abundance of fluorescently labeled proteins in response to various genetic lesions or environmental insults (Laufer *et al*, 2013; Ljosa *et al*, 2013; Chong *et al*, 2015). Proteins localize to regions of the cell where they are required to carry out specific functions, and a change in protein localization following a genetic or environmental perturbation often reflects a critical role of the protein in a biological response of interest. High-throughput (HTP) microscopy enables analysis of proteome-wide changes in protein localization in different conditions, providing data with the spatiotemporal resolution that is needed to understand the dynamics of biological systems.

The budding yeast, *Saccharomyces cerevisiae*, remains a premiere model system for the development of experimental and computational pipelines for HTP phenotypic analysis. A key high-quality resource for yeast imaging experiments is the open reading frame (ORF)-GFP fusion collection (Huh *et al*, 2003) which consists of 4,156 strains, each expressing a unique ORF-GFP fusion gene, whose expression is driven by the endogenous ORF promoter. The GFP-tagged yeast collection has been used to assign 75% of the budding yeast proteome to 22 distinct localizations under standard growth conditions, using manual image inspection. Several studies have since used the collection to quantify protein abundance changes and to map protein re-localization in response to various stress conditions, again using manual assessment of protein localization (Tkach *et al*, 2012; Breker *et al*, 2013).

More recently, efforts have been made to develop computational methods for systematic and quantitative analysis of proteome dynamics in yeast and other cells (Breker & Schuldiner, 2014; Grys *et al*, 2017). For example, our group classified images of single yeast cells from screens of the ORF-GFP collection into one or more of 15 unique subcellular localizations using an ensemble of 60 binary support vector machine (SVM) classifiers. Each SVM classifier was trained on manually annotated sample images of single cells, with a training set containing > 70,000 cells in total. Overall, this classifier ensemble (ensLOC) performed with > 70% precision and recall,

1 Department of Electrical and Computer Engineering, University of Toronto, Toronto, ON, Canada
2 Donnelly Centre for Cellular and Biomolecular Research, University of Toronto, Toronto, ON, Canada
3 Department of Molecular Genetics, University of Toronto, Toronto, ON, Canada
4 Cellular Pharmacology, Discovery Sciences, Janssen Pharmaceutical Companies, Johnson & Johnson, Beerse, Belgium
5 Canadian Institute for Advanced Research, Program on Genetic Networks, Toronto, ON, Canada
6 Canadian Institute for Advanced Research, Program on Learning in Machines & Brains, Toronto, ON, Canada
 *Corresponding author. Tel: +1 416 946 7260; E-mail: charlie.boone@utoronto.ca
 **Corresponding author. Tel: +1 416 978 6113; E-mail: brenda.andrews@utoronto.ca
 † These authors contributed equally to this work

providing a quantitative localization output not achievable using manual assessment (Koh *et al*, 2015). The ensLOC approach also outperformed earlier automated methods also based on SVMs for classifying the ORF-GFP fusion collection (Chen *et al*, 2007; Huh *et al*, 2009).

Attempts to apply the ensLOC classifiers to new microscopy datasets involved a significant amount of re-engineering and supplemental training. This problem reflects limitations associated with the image features used to train the classifiers. Typically, single cells are segmented from the images and hundreds of measurements representing pixel intensity statistics and patterns are computed for each cell (Chen *et al*, 2007; Dénervaud *et al*, 2013; Loo *et al*, 2014; Chong *et al*, 2015; Lu & Moses, 2016). The high dimensional feature space is then reduced by selecting relevant features for the classification task or using dimensionality reduction techniques prior to training a classifier (Liberali *et al*, 2014; Kraus & Frey, 2016). These segmentation and feature reduction techniques are typically not transferable across datasets, thereby requiring researchers to tune and re-train analysis pipelines for each new dataset.

Deep learning methods have the potential to overcome the limitations associated with extracted feature sets by jointly learning optimal feature representations and the classification task directly from pixel level data (LeCun *et al*, 2015). Convolutional neural networks in particular have exceeded human-level accuracy at the classification of modern object recognition benchmarks (He *et al*, 2015) and their use is being adopted by the biological imaging field. Recently, deep learning has been applied to the classification of protein localization in yeast (Kraus *et al*, 2016; Pärnamaa & Parts, 2016), imaging flow cytometry (Eulenberg *et al*, 2016), as well as the classification of aberrant morphology in MFC-7 breast cancer cells (Dürr & Sick, 2016; Kraus *et al*, 2016). In addition, recent publications report that feature representations learned by training convolutional networks on a large dataset can be used to extract useful features for other image recognition tasks (Razavian *et al*, 2014; Pawlowski *et al*, 2016), and that previously trained networks can be updated to classify new datasets with limited training data, a method referred to as "transfer learning" (Yosinski *et al*, 2014).

Here, we demonstrate that the application of deep neural networks to biological image data overcomes the pitfalls associated with conventional machine learning classifiers with respect to performance and transferability to multiple datasets. We offer an open-source implementation capable of updating our pre-trained deep model on new microscopy datasets within hours or less. This model is deployable to entire microscopy screens with GPU or CPU cluster-based acceleration to overcome the significant computational bottleneck in high-content image analysis.

# Results

## Training and validating a deep neural network (DeepLoc) for classifying protein subcellular localization in budding yeast

Toward our goal of building a transferable platform for automated analysis of high-content microscopy data, we constructed a deep convolutional neural network (DeepLoc) to re-analyze the yeast protein localization data generated by Chong *et al* (2015). We provide a brief overview of convolutional neural networks in Fig EV1 and refer readers to LeCun *et al* (2015) and Goodfellow *et al* (2016) for a more thorough introduction. To make a direct comparison of DeepLoc and ensLOC performance, we decided to train our network to identify and distinguish the same 15 subcellular compartments identified using the SVM classifiers (Fig 1A). We implemented and trained a deep convolutional network in Tensor-Flow (Abadi *et al*, 2015), Google's recently released open-source software for machine learning (Rampasek & Goldenberg, 2016). In DeepLoc, input images are processed through convolutional blocks in which trainable sets of filters are applied at different spatial locations, thereby having local connections between layers, and enabling discovery of invariant patterns associated with a particular class (e.g., nucleus or bud neck). Fully connected layers are then used for classification, in which elements in each layer are connected to all elements in the previous layer. Our network arranges 11 layers into eight convolutional blocks and three fully connected layers, consisting of over 10,000,000 trainable parameters in total (more detail in Materials and Methods, network architecture shown in Fig 1B). To ensure the validity of our comparative analysis, we trained DeepLoc on a subset of the exact same manually labeled cells used to train ensLOC (Chong *et al*, 2015), totaling ~22,000 images of single cells. However, instead of training a classifier on feature sets extracted from segmented cells, we trained DeepLoc directly on a defined region of the original microscopy image centered on a single cell, but often containing whole, or partial cells in the periphery of the bounding box. The use of these "bounding boxes" removes the sensitivity of the image analysis to the accuracy of segmentation that is typical of other machine learning classifiers. Despite using a substantially smaller training set than was used to train ensLOC (Chong *et al*, 2015) (~70% fewer cells), we found that training a single deep neural network using a multi-class classification setting substantially outperformed the binary SVM ensemble when assigning single cells to subcellular compartment classes (71.4% improvement in mean average precision, Fig 1C).

The ensLOC method relied on aggregating across cell populations to achieve > 70% precision and recall in comparison with manually assigned protein localizations (Huh *et al*, 2003). To assess the performance of DeepLoc in a similar way, we aggregated cell populations by computing the mean for each localization category across single cells containing the same GFP fusion protein. Again, DeepLoc outperformed the binary classifier ensemble across all localization categories (Fig 1D), achieving a mean average precision score (area under precision recall curve) of 84%, improving on the classification accuracy of ensLOC by almost 15% with substantially less training input.

## Visualizing network features

Having demonstrated the improved performance of DeepLoc over the analysis standard, we next investigated which components of our network were contributing to its success. One of the hallmark differences between deep networks and traditional machine learning is that the network's learned representations are better at distinguishing between output classes than extracted feature representations used by other classifiers. To address whether this difference was relevant in our experiments, we visualized the activations of the final convolutional layer in 2D using t-distributed stochastic neighbor embedding (t-SNE) (Maaten & Hinton, 2008) for a single

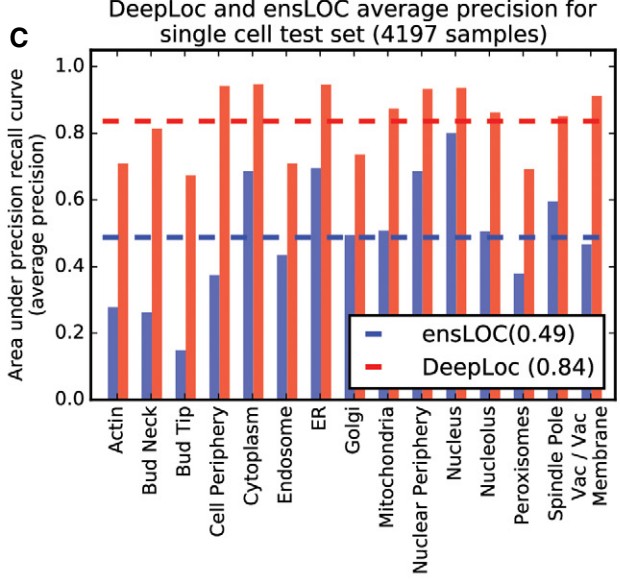

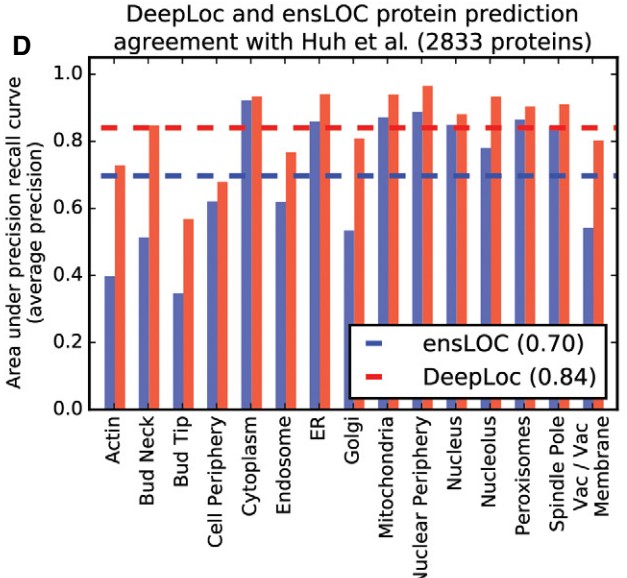

**Figure 1.**

cell test set (Fig 2A). t-SNE is a popular non-linear dimensionality reduction algorithm often used to visualize the structure within high dimensional data in 2D or 3D space. Similarly, we visualized the

CellProfiler (Carpenter *et al*, 2006)-based features used to train the ensLOC SVM ensemble (Chong *et al*, 2015) on the exact same test set of single cell images (Fig 2B). We observed that using the

**Figure 1.    DeepLoc input data, architecture, and performance.**

A    Example micrographs of yeast cells expressing GFP-tagged proteins that localize to the 15 subcellular compartments used to train DeepLoc.

B    Architecture of DeepLoc illustrating the structure of typical convolutional blocks, max pooling, and fully connected layers. The flowchart focuses on a sample image with a GFP fusion protein that localizes to the nuclear periphery (input). The input is processed through a series of repeating convolutional blocks (orange) and max pooling layers (yellow). In the convolutional block, the activation images illustrate network representations of the sample image (input). The red box and dashed/solid lines illustrate the connections within convolutional layers. Max pooling (yellow blocks) down sample activations across spatial dimensions. After repeated processing through convolutional blocks and max pooling, three fully connected layers are used for classification (green). The last layer (output) represents the distribution over localization classes.

C    Average precision of DeepLoc (red bars) and ensLOC (Chong *et al*, 2015) (blue bars) on classifying a single cell test set (*n* = 4,197 samples). The cell compartment is indicated on the *x*-axis and the average precision (area under the precision recall curve) on the *y*-axis. The dashed lines indicate the mean average precision across the localization classes (0.49 for ensLOC (Chong *et al*, 2015) and 0.84 for DeepLoc).

D    Average precision of DeepLoc (red bars) and ensLOC (Chong *et al*, 2015) (blue bars) on assigning localizations to images of GFP fusion proteins with single or multiple localization classes according to manual annotations by Huh *et al* (2003) (*n* = 2,833 proteins). The cell compartment is indicated on the *x*-axis and the average precision (area under the precision recall curve) on the *y*-axis. The dashed lines indicate the mean average precision across the localization classes (0.70 for ensLOC (Chong *et al*, 2015) and 0.84 for DeepLoc).

DeepLoc representations, cells appeared to be better arranged in accordance with their localization classes, suggesting that DeepLoc's convolutional layers learn to extract features that are meaningful in the distinction of protein subcellular localization. These results suggest that an important component of the improved performance of DeepLoc reflects the network's ability to learn feature representations optimized directly on pixel values for a specific classification task as opposed to training classifiers on static feature sets.

Next, we wanted to display these features to assess how they differ between compartment classes. To do this, we visualized activations and patterns extracted in the last convolutional layer of the network (layer 8) for specific input examples (Golgi, bud neck, nuclear periphery, Fig 2C, Materials and Methods). Different input patterns activated specific features in deeper convolutional layers (convolutional activations, Fig 2C), with representations being combined in the fully connected layers from the convolutional feature maps, ultimately producing unique signals for different input patterns. These signals differ by localization class in a biologically interpretable way. For example, images containing punctate subcellular structures like the Golgi (top panels, Fig 2C) activated similarly patchy, dispersed features, while images containing discrete compartments like the bud neck (middle panels, Fig 2C) activated features that appear localized and linear.

We extended our analysis by applying activation maximization (Yosinski *et al*, 2015) to visualize input patterns that maximally activate each output class (Fig 2D, see Materials and Methods). This technique works by keeping the parameters of the network constant while updating input pixel values to maximize the activation of specific features. In our implementation, the network iteratively updates an input with a randomly initialized green channel to produce an example "input" that resembles a cell with a GFP fusion protein that localizes to the maximally activated output class. The visualizations produced by the network for different output categories were convincing in their similarity to real compartment architecture. For example, visualizations for compartments such as the actin cytoskeleton, peroxisomes, and the spindle pole body were all punctate and dispersed (Fig 2D). Although these general visualizations may place compartments in various locations in the cell due to variable compartment locations in different images (e.g., spindle pole), the general morphology remains biologically interpretable. These results further justify the use of deep learning for classifying protein subcellular localization.

## Using DeepLoc to identify protein dynamics in response to mating pheromone

Next, we assessed the ability of DeepLoc to classify images of yeast cells generated in different microscopy screens from those that served as training input to the network. We opted to analyze images from a screen generated by our group at the same time and on the same HTP confocal microscope as our previously published wild-type screens (Chong *et al*, 2015), but that ensLOC had been unable to accurately classify. In this genome-wide screen, haploid *MAT***a** cells were exposed to the mating pheromone α-factor, causing cell cycle arrest in G1 phase and polarized growth of a mating projection (schmoo) (Merlini *et al*, 2013). We used DeepLoc to analyze 16,596 images of the ORF-GFP collection acquired after exposure to mating pheromone for 40, 80, and 120 min. Images and analysis are available on the Cyclops Database (http://cyclops.ccbr.utoronto.ca). We reasoned that a pheromone response time course would be a challenging test case for DeepLoc, due to the dramatic changes in cell morphology associated with α-factor treatment. DeepLoc produced reasonable protein classifications for single cells within hours, without the need for additional, non-wild-type training, while re-implementing an SVM ensemble would have necessitated weeks of training and optimization.

We identified 297 proteins (Table EV1) whose localization changed significantly in response to α-factor using Welch's *t*-test to score localization changes and a mixture model to identify significance (see Materials and Methods). The 100 proteins demonstrating the most substantial localization changes were significantly enriched for proteins with annotated roles in conjugation and sexual reproduction (Gene Ontology bioprocess; $P < 0.01$). This subset was also enriched for proteins required for cell fusion (e.g., Fus1, Fus2, Fus3, $P < 0.01$), nuclear fusion during mating (e.g., Prm3, Fig2, Kar5, $P < 0.01$), and polarized growth of the mating projection (e.g., Bni1, Pea2, Cdc24, $P < 0.05$). DeepLoc's ability to identify the movement of proteins that are already implicated in the mating response program serves to validate our method for detecting biologically meaningful results.

To do this, in addition to the localization measurements calculated by DeepLoc, we also extracted pixel intensity measurements as a metric for protein abundance (Tkach *et al*, 2012; Breker *et al*, 2013; Chong *et al*, 2015) (Table EV2). In total, we detected 82 proteins whose abundance changed 2-fold or more in response to

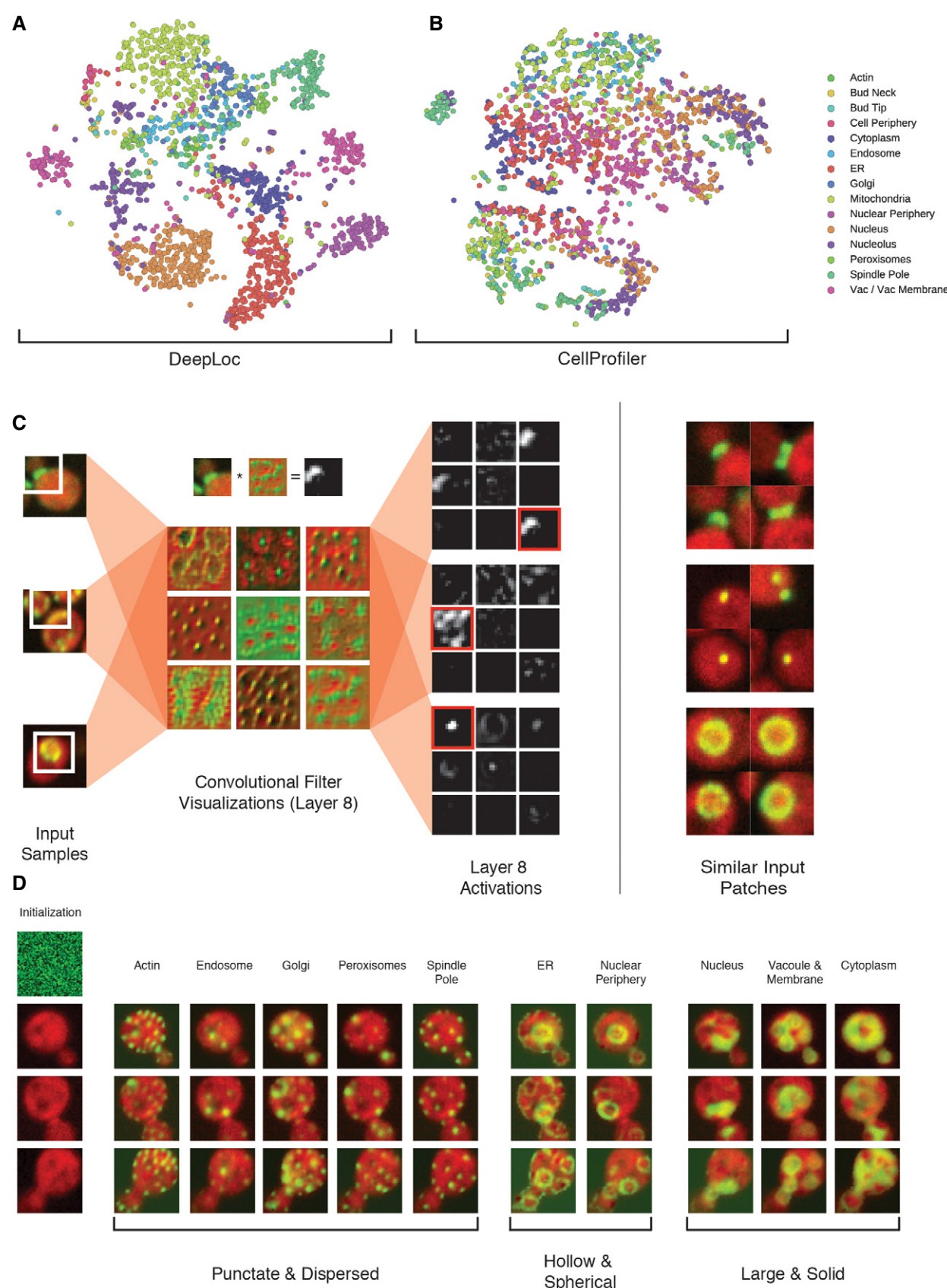

**Figure 2.**

**Figure 2.  Visualizing DeepLoc features.**

A  2D t-SNE (Maaten & Hinton, 2008) visualization of activations in the last convolutional layer of DeepLoc for 2,103 single cells in the test set. We computed the maximum activation across the spatial coordinates for each of the 256 features prior to fitting t-SNE.

B  t-SNE visualization of CellProfiler features extracted for the same cells. We normalized the 313 CellProfiler features to be in the range [0,1]. In these plots, each circle represents a single cell; circles are colored by their localization as determined by manual annotation (Huh *et al*, 2003) (color code to the right).

C  Filters and activations in the last convolutional layer of DeepLoc for sample input images containing GFP fusion proteins that localize to the bud neck (top), Golgi (middle), or nuclear periphery (bottom). The convolutional filter visualizations were generated by activation maximization (Yosinski *et al*, 2015). The maximally activated filter for each input is highlighted with a red box (bud neck at the top, Golgi in the middle, and nuclear periphery at the bottom). For the bud neck sample, the input patch, filter, and activation are presented together to visualize how features are activated in DeepLoc. Other input patches that also maximally activate the selected feature are displayed.

D  Regularized activation maximization (Yosinski *et al*, 2015) of output layers based on inputs initialized to leftmost column (Initialization). Different localization classes (compartment labels at the top of the images) are grouped by their morphological similarity (labels at bottom of images).

pheromone, with 75 proteins increasing in abundance and seven proteins decreasing in abundance. Although there are minimal data available for protein abundance changes in α-factor, we compared our abundance measurements to gene expression changes and found positive correlations that are largely driven by the strongest hits (Fig EV2). While unrelated to the localization analysis by DeepLoc, this evaluation of protein abundance further validates the effectiveness of our screening protocol; it also provides a complementary overview of proteomic responses to those made by Chong *et al* (2015) in the Cyclops database.

Next, we wanted to display a quantitative snapshot of these proteomic responses to α-factor treatment similar to those previously constructed to illustrate protein movement after treatment with rapamycin, hydroxyurea, or the deletion of *RPD3* (Chong *et al*, 2015). We displayed proteins with the most substantial localization changes (*t*-test statistic with magnitude > 10) in a flux network, indicating if these proteins changed in abundance as well (Fig 3A). As previously reported (Chong *et al*, 2015), after exposure to an environmental perturbation, we observe that proteins change in abundance or localization but rarely in both. Representative micrographs illustrate interesting localization/abundance changes shown in the flux network (Fig 3B). Importantly, DeepLoc identified novel movements of proteins already implicated in the mating response, such as the movement of Kss1, a MAPK that functions primarily to regulate filamentous growth, from the nucleus to the cytoplasm. We also identified the appearance of cell fusion regulators Prm1, Prm2, and Fus1 at the vacuole, which presumably results from the endocytosis of these cell surface proteins. Importantly, DeepLoc also identified the known localization of Prm1 and Prm2 at the Schmoo/bud tip (Heiman & Walter, 2000), though this movement is not shown on the flux network as their localization at the vacuole is more substantial. Deeploc also identified changes in localization of a number of proteins that control bud site selection, including Bud2, Bud4, and Bud5, which presumably reflects the fact that pheromone signaling is controlling polarized growth and over-riding the bud site selection machinery.

In addition to these striking changes, DeepLoc also identified more subtle or partial localization changes. For example, Nvj1 localized primarily to the spindle pole in untreated cells, but was also present at the nuclear periphery, as previously reported, where it performs a role in the formation of nucleus-vacuole junctions (Pan *et al*, 2000). After treatment with α-factor, DeepLoc captured Nvj1's movement away from the spindle pole, and its enhanced localization at the nuclear periphery. A number of proteins with no or poorly annotated roles also show clear localization changes,

implicating these proteins in the pheromone response. For example, an uncharacterized protein Yor342c moved from the nucleus to the cytoplasm after α-factor treatment, a relocalization that has been previously noted in response to DNA replication stress (Tkach *et al*, 2012).

## Assessing the transferability of DeepLoc to new and different microscopy datasets

With the goal of generating an automated image analysis system that can be broadly implemented by the budding yeast community, we used transfer learning (Yosinski *et al*, 2014) to classify image sets that significantly diverge from the images used to train DeepLoc. First, we completed a new genome-wide screen in standard cell culture conditions, which we called wild-type (WT)-2017, using the budding yeast ORF-GFP fusion collection (Huh *et al*, 2003). To differentiate this image set from other datasets analyzed by DeepLoc, screens were performed using a new HTP confocal microscope, and strains contained different red fluorescent markers (See Materials and Methods, cropped cell images available at: http://spidey.ccbr.utoronto.ca/~okraus/DeepLoc_full_datasets. zip). We incorporated five new localization classes, many of which are punctate (e.g., Cytoplasmic foci, eisosomes, and lipid particles) and likely difficult to differentiate using traditional machine learning approaches, explaining their absence from ensLOC (localization classes shown in Fig 4A). We transferred and fine-tuned DeepLoc to the WT-2017 dataset using an increasing amount of training input per class, and contrasted the performance of this network with one trained from scratch using the same amount of training input (See Materials and Methods; Fig 4B). Remarkably, transfer learning using DeepLoc achieved an average accuracy of 62.7% when fine-tuned with only five additional supplemental training cells per class (Fig 4C, yellow highlight), with several localization categories achieving accuracies above 80% (Fig 4D); this is a 63.4% improvement in performance using transfer learning over training from scratch (Fig 4E). The classes with significant errors are mostly the new punctate localizations, including cytoplasmic foci, and lipid particles, which are difficult to differentiate with only a few samples, and are still identified with 63.8% accuracy when merged with peroxisomes into one class.

Next, we used our transfer learning protocol to classify images generated by the Schuldiner laboratory using a different microscope and fluorescent markers (Yofe *et al*, 2016). Because these images were never intended for automated analysis, they contain many cells that are often clustered and overlapping. Also, bright field

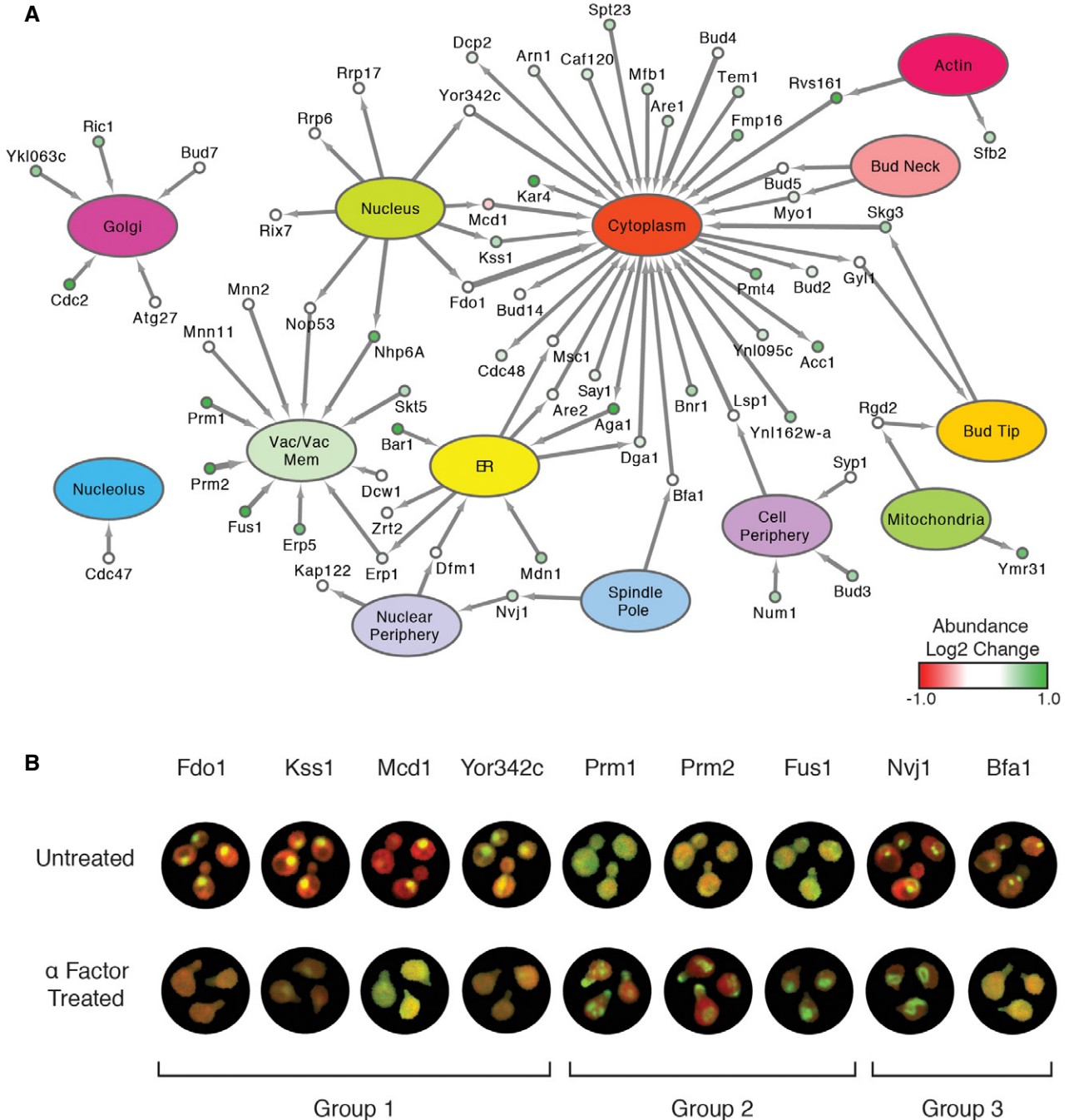

**Figure 3.  Protein dynamics in response to mating pheromone.**

A   Flux network (Chong *et al*, 2015) showing significant protein localization and abundance changes in response to the mating pheromone α-factor. Localization changes with *t*-scores above 10 are shown. Hubs represent cellular compartments, while nodes represent proteins. Nodes are colored to represent abundance changes for those proteins that are changing in both their localization as well as abundance. Edge thickness corresponds to the magnitude of the localization change score.

B   Representative micrographs highlighting protein subcellular movements after treatment with α-factor. Group 1: proteins that move from the nucleus to the cytoplasm. Group 2: proteins that appear in the vacuole/vacuolar membrane. Group 3: proteins that are moving away from the spindle pole after treatment with α-factor.

imaging was used to identify outlines of the cells, which do not express a fluorescent cytosolic marker (Fig 5A). Despite these significant differences, we were able to use transfer learning with DeepLoc (Fig 5B) to classify protein localizations in this dataset with an average accuracy of 63.0% after training with only 100 samples per class (Fig 5C). Classification accuracy with transfer

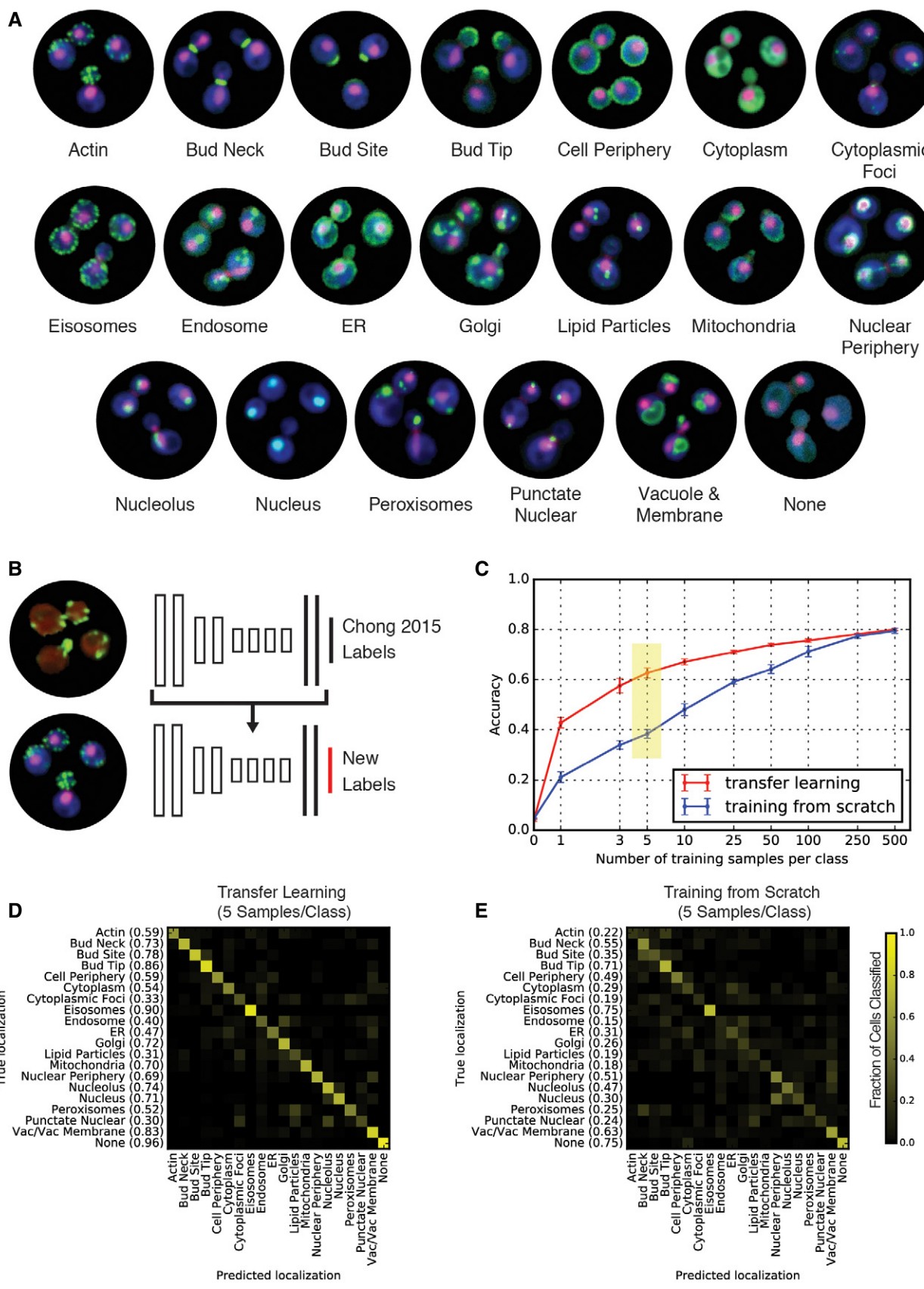

Figure 4.

**Figure 4. Performance of DeepLoc after transfer learning.**

A Example micrographs from a screen of wild-type yeast cells expressing ORF-GFP fusion proteins. The images are of single cells expressing fusion proteins that localize to 20 unique output classes (colored green). The cells also express a bright cytosolic marker (FarRed; colored blue), as well as a nuclear RFP fusion protein (colored red).

B Illustration of transfer learning. All layers except for the last layer (in red) are initialized to the network trained on the Chong *et al* (2015) dataset.

C Comparison of classification accuracy (*y*-axis) for different training set sizes (*x*-axis) when transfer learning is implemented using DeepLoc (red line) versus training a network from scratch (blue line). Error bars indicate the standard deviation of the accuracy based on five different samplings of the training set for each training set size. A yellow box highlights network versions that are referred to in (D and E).

D Confusion matrix for transfer learning the DeepLoc network trained on the Chong *et al* (2015) dataset to the new dataset with five samples per class. The intensity of the yellow color in each block of the matrix indicates the fraction of cells classified from each class predicted to be in a given class (scale bar to the right). Prediction accuracy for each class is indicated in brackets on the *y*-axis.

E Confusion matrix for training DeepLoc from random initializations with five samples per class.

learning ranged from 79% for the mitochondrial and "punctate" compartments to 41% for the bud compartment (Fig 5D). The availability of unique cell images for training varied by localization class, which likely affected accuracy in some cases (see Materials and Methods, Table EV3). In contrast, performance was reduced for all classes when DeepLoc was trained from scratch (Fig 5E). Despite these classification errors, the performance of DeepLoc is a significant achievement given that these images have previously only been classified by manual inspection, and that the imaging protocols were highly divergent from those that are optimized for automated analysis.

# Discussion

We describe the implementation of a deep neural network for automated analysis of HTP biological images. DeepLoc overcomes several limitations associated with existing machine learning pipelines for computational classification of cell morphologies and phenotypes. Machine learning models that have been used to cluster or classify individual cells using selected features, although useful (Stadler *et al*, 2013; Chong *et al*, 2015; Styles *et al*, 2016), often fail to classify complex phenotypes. For example, using ensLOC (Chong *et al*, 2015), 35% of vacuolar proteins were assigned to the nucleus and proteins localizing to punctate compartments including actin patches, spindle pole bodies, peroxisomes, and nucleoli were occasionally mis-classified. For these cases, poor classification performance can be attributed to the limited space of patterns that can be represented by static feature sets that are extracted prior to classifier training. In contrast, DeepLoc updates its parameters by training directly on image data, thereby enabling it to learn patterns optimized for the classification task. DeepLoc achieves a 47% increase

in average precision for proteins with vacuolar localizations compared to ensLOC (Chong *et al*, 2015; Fig 1D).

An additional limitation of traditional machine learning approaches is that each step in the analysis pipeline needs to be tuned for different experiments, severely limiting their utility for cell biologists. The machine learning algorithms typically used, including SVMs and clustering algorithms, are sensitive to segmentation, data preprocessing, and feature selection steps. DeepLoc overcomes these limitations by training directly on bounding boxes surrounding single cells. The lack of dependence on accurate segmentation and the large variety of patterns that can be learned from the large training set enabled DeepLoc to accurately classify cells in challenging datasets where cell morphology is abnormal, such as yeast cells treated with pheromone (Fig 3B). Furthermore, this feature enabled DeepLoc to analyze images generated through a highly divergent microscopy screen performed by another laboratory with limited additional training. In this case, transfer learning from DeepLoc achieved 72.3% accuracy at classifying images that were not optimized for automated analysis, an attribute that is a prerequisite for ensuring that analysis pipelines are broadly applicable to the cell biology community (Fig 5C).

These results differentiate DeepLoc from previous implementations of deep learning for high-throughput cell image data. Recent publications demonstrate the improved accuracy achieved by deep learning-based classifiers for high-content screening (Dürr & Sick, 2016; Kraus *et al*, 2016; Pärnamaa & Parts, 2016) and for imaging flow cytometry (Eulenberg *et al*, 2016). These reports validate their proposed models on held out test sets from the same source as the training data and typically evaluate less phenotypes than DeepLoc (i.e., four mechanism of action clusters in Dürr and Sick (2016) and five cell cycle stages in Eulenberg *et al* (2016)). In Kraus *et al* (2016), we describe a deep learning framework for classifying whole

**Figure 5. Performance of DeepLoc for classifying images of cells expressing ORF-RFP fusion proteins collected for manual assessment.**

A Example micrographs from a screen of wild-type yeast cells expressing ORF-RFP fusion proteins (Yofe *et al*, 2016). The images are of single cells expressing ORF-RFP fusion proteins that localize to 10 unique output classes. The cells express a single RFP fusion protein of interest; cell outlines are visualized in brightfield.

B Illustration of transfer learning. All layers except for the last layer (in red) are initialized to the network trained on the Chong *et al* (2015) dataset.

C Comparison of classification accuracy (*y*-axis) for different training set sizes (*x*-axis) when transfer learning is implemented using DeepLoc (red line) versus training a network from scratch (blue line). Error bars indicate the standard deviation of the accuracy based on five different samplings of the training set for each training set size. A yellow box highlights network versions that are referred to in (D and E).

D Confusion matrix for transfer learning the DeepLoc network trained on the Chong *et al* (2015) dataset to the new dataset with 100 samples per class. The intensity of the yellow color in each block of the matrix indicates the fraction of cells classified from each class predicted to be in a given class (scale bar to the right). Prediction accuracy for each class is indicated in brackets on the *y*-axis.

E Confusion matrix for training DeepLoc from random initializations with 100 samples per class.

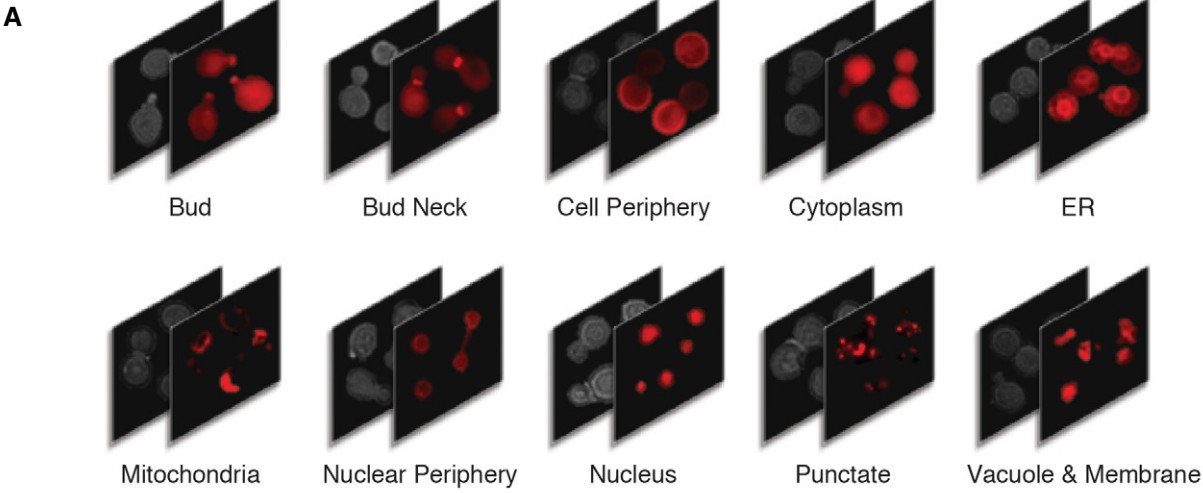

**A**

Bud          Bud Neck          Cell Periphery          Cytoplasm          ER

Mitochondria          Nuclear Periphery          Nucleus          Punctate          Vacuole & Membrane

**B**

Chong 2015 Labels

New Labels

**C**

- transfer learning
- training from scratch

Accuracy

Number of training samples per class

**D**          Transfer Learning
(100 Samples/Class)

True localization

Bud (0.41)
Bud Neck (0.43)
Cell Periphery (0.72)
Cytoplasm (0.72)
ER (0.70)
Mitochondria (0.79)
Nuclear Periphery (0.56)
Nucleus (0.57)
Punctate (0.79)
Vac/Vac Membrane (0.55)

Predicted localization

**E**          Training from Scratch
(100 Samples/Class)

True localization

Bud (0.38)
Bud Neck (0.21)
Cell Periphery (0.26)
Cytoplasm (0.40)
ER (0.46)
Mitochondria (0.41)
Nuclear Periphery (0.26)
Nucleus (0.34)
Punctate (0.42)
Vac/Vac Membrane (0.39)

Predicted localization

Fraction of Cells Classified

**Figure 5.**

microscopy images that is not designed to classify single cells. Here, we train DeepLoc on 15 subcellular localizations classes from one genome-wide screen, deploy DeepLoc to a second genome-wide screen of cells with substantially altered cell morphology that was not amenable to classification with EnsLoc, and then use transfer learning to deploy DeepLoc to image sets that were screened differently than the training set with minimal additional labeling.

Based on our findings, we believe that deep learning models as well as corresponding image sets should be shared across the high-content screening community. The feature representations learned by these models will only become more powerful as they are trained on varied datasets and on different classification tasks. The machine learning community has numerous platforms for sharing models (Caffe Model Zoo, 2016, Models built with TensorFlow, 2016) and datasets (Russakovsky *et al*, 2015), and within the high-content screening community, the Broad Institute has provided a repository for microscopy images (Ljosa *et al*, 2012). Here, we provide our trained models and training sets with the goal of encouraging others to adopt and share deep learning approaches for characterizing high-content microscopy screens. Based on our implementation of transfer learning from DeepLoc, the use of this technology will aid many groups in the analysis of their image-based data, ultimately expediting future discoveries in the high-content screening community.

# Materials and Methods

### Strain construction for α-factor and WT-2017 datasets

*Saccharomyces cerevisiae* strains were generated by first constructing *MATα* query strains with red and far-red fluorescent markers using standard PCR and yeast transformation by homologous recombination and then crossing these queries into the ORF-GFP fusion array using a modified SGA protocol. For the α-factor dataset, our query contained a single red fluorescent marker (*RPL39pr-tdTomato*) of the cytosol, while the WT-2017 dataset contained a far-red cytosolic marker (*TDH3pr-E2crimson*), as well as a red fluorescent marker (*HTA2-mCherry*) of the nucleus and a second red fluorescent marker (*CDC11-TagRFP*) of the bud neck. Strain genotypes are:

**α-Factor query (BY4872):** *MATα hoΔ::NAT CAN1pr::RPL39pr-tdTomato::CaURA3::can1Δ::STE2pr-LEU2 lyp1Δ ura3Δ0 his3Δ1 leu2Δ0 met15Δ0*

**WT-2017 query (BY5282):** *MATα CAN1pr::TDH3pr-E2Crimson::HPH::can1Δ::STE2pr-LEU2 HTA2-mCherry::CaURA3 CDC11-TagRFP::NAT lyp1Δ ura3Δ0 his3Δ1 leu2Δ0 met15Δ0*

### Strain preparation for imaging and growth conditions

Haploid *MAT**a*** strains were inoculated into liquid growth medium, diluted, and prepared for imaging using slightly different protocols for each dataset; details are provided below.

#### α-Factor dataset

Haploid *MAT**a*** strains were inoculated into synthetic medium with 1 mM methionine, 100 μg/ml NAT, and 2% glucose. Cultures were grown to saturation in 96-well microplates (200 μl volume) with glass beads. Next, cultures were diluted in low fluorescence

synthetic medium containing 5 g/l L-glutamic acid monosodium salt hydrate (MSG), 1 mM methionine, 100 μg/ml NAT, and 2% glucose, in 600 μl deep well blocks with glass beads. Cultures were grown to early log phase and then transferred to 384-well glass-bottom imaging plates (CellCarrier, Perkin Elmer) using a Zymark RapidPlate liquid handling device. Cultures were exposed to 5 μM α-factor for 2 h prior to image acquisition.

#### WT-2017 dataset

Haploid *MAT**a*** strains were inoculated into synthetic medium with 1 mM methionine, 100 μg/ml NAT, 100 μg/ml hygromycin B, and 2% glucose. Cultures were grown to saturation in 96-well microplates (200 μl volume) with glass beads. Cultures were diluted in low fluorescence synthetic medium containing 5 g/l ammonium sulfate, 1 mM methionine, 20 μg/ml ampicillin, and 2% glucose, in 600 μl deep well blocks with glass beads. Cultures were grown to early log phase and then transferred to 384-well glass-bottom imaging plates (CellCarrier, Perkin Elmer) using a Zymark RapidPlate liquid handling device.

### Live-cell image acquisition

Liquid cultures were imaged by slightly different acquisition protocols on different HTP confocal microscopes for each dataset; details are provided below.

#### α-Factor dataset

Images were acquired using an HTP spinning-disk confocal microscope (Opera, PerkinElmer) with a 60× water-immersion objective (NA 1.2, image depth 0.6 μm and lateral resolution 0.28 μm). Sample excitation was conducted using two lasers (488 and 561 nm) at maximum power for 800 ms exposures per site. Two cameras (12-bit CCD) were used to simultaneously acquire images of red and green fluorescence after excitation (binning = 1, focus height = 2 μm). Briefly, a 405/488/561/640 nm primary dichroic and a 564 nm detector dichroic were used to passage light toward both cameras. A 520/35-nm filter was placed in front of camera 1 for the separation of green fluorescence, while a 600/40-nm filter was placed in front of camera 2 for the separation of red fluorescence. A total of four images were acquired in each channel (1,349 × 1,004 pixels), resulting in a total screening time of ~40 min per 384-well plate.

#### WT-2017 dataset

Images were acquired using a different HTP spinning-disk confocal microscope of the same model (Opera, PerkinElmer), also with a 60× water-immersion objective (NA 1.2, image depth 0.6 μm and lateral resolution 0.28 μm). Sample excitation was conducted using three lasers (488, 561, and 640 nm) at maximum power for 800 ms exposures per site. Three cameras (12-bit CCD) were used to simultaneously acquire images of far-red, red and green fluorescence after excitation (binning = 1, focus height = 2 μm). Briefly, a 405/488/561/640 nm primary dichroic and a 564 nm detector dichroic were used to passage light toward both cameras. A 520/35-nm filter was placed in front of camera 1 for the separation of green fluorescence, while a 585/20-nm filter was placed in front of camera 2 for the separation of red fluorescence, and a 690/70-nm filter was placed in front of camera 3 for the separation of far-red fluorescence. A

total of 10 images (1,338 × 1,003 pixels) were acquired in each channel, resulting in a total screening time of ~100 min per 384-well plate.

### Training DeepLoc

We trained a deep convolutional model with 11 layers consisting of eight convolutional blocks and three fully connected layers (Fig 1B). The convolutional layers included convolution with 3 × 3 filters using a stride of 1 followed by the addition of a bias term and a rectified linear unit activation function. The numbers of feature maps in the convolutional layers were 64, 64, 128, 128, 256, 256, 256, and 256. Max pooling was applied with a window size of 2 and stride of 2 after convolutional layers 2, 4, and 8. Following the third pooling layer, activations in the feature maps were flattened into a vector and subsequent layers were fully connected. We used three fully connected layers with 512, 512, and 19 features. We applied batch normalization (Ioffe & Szegedy, 2015) at every layer prior to applying the rectified linear unit activation function. The last layer represents the localization classes in the trainings set. We applied the softmax function to the activations of the final output layer to produce a distribution over the localization classes. In total, the network has over 10,000,000 trainable parameters. We implemented and trained the network in TensorFlow (Abadi *et al*, 2015) using the Adam optimization algorithm (Kingma & Ba, 2014). Network parameters were initialized using a truncated normal distribution function with a standard deviation of 0.1. We trained the network for 10,000 iterations using a batch size of 128 and a learning rate with an exponential decay of 0.96 applied every 25 iterations, starting from a value of 0.1. Following training, we evaluated the validation performance of the network on models saved at iteration intervals of 500 and chose the best performing model for subsequent evaluations.

We trained the network using 21,882 single cells that had previously been manually assigned to one of 17 localization compartments by Chong *et al* (2015) (including two quality control classes for identifying dead cells and inputs without cells; Fig 1A). The original labeled dataset was composed of 60 subdatasets, each containing "positive" and "negative" samples, to train the 60 binary SVM classifiers used in ensLOC. Instead of using all of the > 70,000 cells previously annotated, we sampled only positive examples such that each localization compartment contained 185–1,500 cells (see Table EV3 for exact numbers per class) and we trained DeepLoc as a multi-class classifier. We obtained the *x, y* coordinates for each training sample from the previous features sets extracted using CellProfiler (Carpenter *et al*, 2006). For each cell, we cropped a bounding of 64 × 64 pixels centered on its *x, y* coordinates. Yeast cells change in their size over cell cycle progression but average 49 pixels along the major axis, and 37 pixels along the minor axis of the cell. Similarly, we extracted validation and test sets consisting 4,516 cells each. To enable the network to classify proteins with various intensity values, we normalized the pixel values of each input cell to be in the range [0,1] by saturating the top and bottom 0.1 percentile of pixel values. To enhance the networks generalization performance, we applied commonly used augmentation and normalization operations to the input samples during training. Specifically, we extracted random 60 × 60 patches from the input samples and trained the network on random vertical and horizontal

reflections of the patches as well as random 90° rotations (Krizhevsky *et al*, 2012). These augmentations help prevent the network from overfitting to the training data while preserving the label of the input samples (as the protein localization patterns are invariant to rotations and reflections). When evaluating DeepLoc on new cells, we computed the mean of predictions produced by the center 60 × 60 crop and 60 × 60 crops extracted from the four corners.

### Evaluating DeepLoc performance

To evaluate single cell performance, we used DeepLoc to produce predictions for every cell in our test set and we regenerated the ensLOC predictions described in Chong *et al* (2015) using the Cyclops database (Koh *et al*, 2015) for the exact same test set. DeepLoc produces a vector representing the distribution over localization classes and ensLOC produces a list of annotations for each single cell. We used these predications to calculate the average precision (area under the precision recall curve) for the two techniques.

We used DeepLoc to produce predictions for every cell in the wild-type screen by obtaining the *x, y* coordinates of each cell from the previously extracted CellProfiler feature sets and cropping the cells as described above. To aggregate the localization predictions for each well, we computed the mean for each localization category across the cells in the well. To compare with Chong *et al* (2015), we used the values reported in the WT1 sheet of Table S2 in their publication. We compared these localization predictions with manually assigned protein localizations (Huh *et al*, 2003; Fig 1D).

### Visualizing network features

To visualize feature representations learned by DeepLoc, we used the t-SNE (Maaten & Hinton, 2008) implementation in Scikit-learn (Pedregosa *et al*, 2012) to visualize the activations in the 8[th] convolutional layer for 2,103 single cells in our test set (Fig 2A). To remove the spatial dependence of these activations, we computed the maximum activation in each feature across the spatial dimensions. Similarly, we visualized the normalized 313 features extracted from CellProfiler (Carpenter *et al*, 2006) for the same set (Fig 2B).

We visualized network activations and patterns activated by specific features as described in the Results. Activation maximization (Yosinski *et al*, 2015) was used to generate input patterns that are maximally activated by network features. This technique works by keeping the parameters of the network constant while updating input pixel values to maximize the activation of specific features using gradient ascent. The input was initialized to uniform noise and iteratively updated by gradient ascent with respect to a specific feature in the network. We used the naïve version of this approach to visualize the last convolutional layer features (Fig 2C). To make the final layer visualizations more interpretable (Fig 2D), we included regularization techniques described in Yosinski *et al* (2015). Specifically, we modified the gradient update with L2 weight decay, used Gaussian blur to smooth high frequency gradients, and clipped gradients with small contributions to the visualization. Additionally, we modified the implementation to produce realistic looking cells by clamping the red cytosolic channel to a specific cell

while only updating the green channel and masked any updates outside the area of the cell.

## Identifying significant localization and abundance changes in α-factor

We used DeepLoc to evaluate images of single cells after treatment with the mating pheromone α-factor screened using the method described above for untreated cells. To identify significant localization changes between the untreated and α-factor screens, we used the test statistic from Welch's *t*-test to obtain a score for each localization class. Next, we identified significant localization changes for each category by fitting a mixture model consisting of a Gaussian distribution and a uniform distribution to these scores. The Gaussian distribution models the background scores, while the uniform distribution is set to have a prior distribution of 1% and models the outlier scores. Scores that were more likely under the outlier distribution were considered to correspond to proteins with significant localization changes between wild-type and α-factor conditions. For some of these proteins, the dominant localization was the same in both conditions although the distributions of localization scores differed significantly.

We used Cytoscape (Shannon *et al*, 2003) to generate a flux network (Fig 3A) after filtering out scores with a magnitude below 10. To identify proteins that changed significantly in abundance between the two screens, we extracted the mean intensity by area within each cell from the extracted CellProfiler (Carpenter *et al*, 2006) feature sets. For each protein, we computed the mean abundance ($I_g$) across the population in both screens. We scored abundance changes for each protein by computing the fold change ($\partial PL$) of the α-factor abundance over the wild-type (untreated) abundance (Chong *et al*, 2015) $\delta PL_{\alpha wt} = log2(I_{g\alpha}/I_{gwt})$. Proteins with a fold change magnitude above 1 ($|\partial PL| > 1$) in at least one time-point following exposure to α-factor (82 in total) were considered to exhibit significant abundance changes (Table EV2).

## Transfer learning

To evaluate the performance of DeepLoc on datasets generated using different strains and microscopes, we implemented a transfer learning protocol. We extracted *x*, *y* coordinates of single cells using a custom segmentation algorithm that uses a mixture of t-distributions to identify nuclei and cytoplasmic regions (Nguyen & Wu, 2012) and then the seeded watershed algorithm (Meyer & Beucher, 1990) to identify individual cells. Individual cells were cropped as described above and were initially labeled according to their protein level annotations. We manually filtered individually cropped cells that were mislabeled using an efficient custom interface that displays sets of 64 cropped cells and allows the user to click on cells to either keep or discard. We used the filtered set of single cells to build training and test sets containing 16,305 and 1,824 cells, respectively (Fig 4A, exact dataset sizes in Table EV3).

We transferred and fine-tuned DeepLoc to this dataset using increasing numbers of examples per class and contrasted the performance of this network with one that had been trained from scratch using the same amount of training input (Fig 4B). We set

up different datasets with 1, 3, 5, 10, 25, 50, 100, 250, and 500 samples from each class. We generated five distinct training sets for each dataset size by sampling from the entire training set randomly. To fine-tune DeepLoc, we initialized a new network in TensorFlow (Abadi *et al*, 2015) with a final layer that corresponds to the new localization categories and we loaded the previously trained parameters from the Chong *et al* (2015) dataset for every layer except the final layer. We applied dropout (Srivastava *et al*, 2014) to the final layer and augmented with training data as described above to prevent the network from overfitting. For each dataset, we updated the network by optimizing the parameters with the Adam optimization algorithm (Kingma & Ba, 2014) using a learning rate of 0.003 for at least 500 iterations with a batch size of 128. In addition, we used the DeepLoc as a feature extractor without additional training by using the activations in the second last fully connected layer as input to several linear classifiers including one-vs-one linear SVM, k-nearest neighbors, random forest, and a fine-tuned version of DeepLoc in which convolutional layers were not updated (Fig EV3). We found that updating all the DeepLoc parameters during transfer learning performed best for all the training set sized used.

We optimized DeepLoc for analysis of images of an ORF-RFP fusion collection generated by a different laboratory (Yofe *et al*, 2016) and lacking markers useful for automated analysis. For this dataset, we used a custom script based on the watershed transform (Meyer & Beucher, 1990) to identify regions in the red channel containing tagged proteins. We used the centroid of each segmented region as a coordinate to center a bounding box around. As before, cropped regions that were initially mis-labeled were filtered out, and the filtered set was used to generate training and test sets consisting of 6,475 and 725 single cells, respectively. Within this training set, some classes including bud, bud neck, and mitochondria have fewer than 500 unique single cell samples (Table EV3). For these classes, we sampled with replacement when the per-class training set size was larger than the number of available unique samples. We applied the same fine-tuning procedure described above.

## Data availability

The image datasets used to train DeepLoc are available for download at: http://spidey.ccbr.utoronto.ca/ ~ okraus/DeepLoc_full_data sets.zip.

The DeepLoc model pre-trained on the Chong *et al* (2015) dataset is available for download at: http://spidey.ccbr.utoronto.ca/ ~ okra us/pretrained_DeepLoc.zip.

A bash script to download and extract both zip files is provided in Code EV1.

The results and images for the α-factor screen are available on the Cyclops Database. Here, individual proteins can be queried using the search function, after which corresponding localization and abundance data from our analysis can be accessed under the "DeepLoc" subheading. Under this subheading, the data from our three untreated conditions (WT1, WT2, and WT3) as well as the three α-factor time-points (AF100, AF140, and AF180) are available for both localization and abundance. In addition, individual micrographs can be accessed under the "Retrieve micrographs from other screen" tab, by selecting "AF100", "AF140", or "AF180": (http://cyc

lops.ccbr.utoronto.ca). Raw images will be made available upon request.

## Code availability

The code for performing the experiments is available for download in Code EV1. We recommend using the up-to-date version available at: https://github.com/okraus/DeepLoc.

**Expanded View** for this article is available online.

## Acknowledgements

This work was supported primarily by Foundation grants FDN-143264 and FDN-143265 from the Canadian Institutes for Health Research and by a grant from the National Institutes of Health (R01HG005853) to B.A. and C.B.. O.Z.K. was supported in part by a Connaught Global Challenge Fund grant to B.A., C.B., and B.F. (University of Toronto) and by the Department of Electrical and Computer Engineering. B.T.G and O.Z.K. hold Ontario Graduate Scholarships. Infrastructure for high-content imaging was acquired with funds from the Canadian Foundation for Innovation and the Ministry of Research and Innovation (Ontario). B.A., C.B., and B.F. are senior fellows and co-director (C.B.) of the Genetic Networks program of the Canadian Institute for Advanced Research, and B.F. is a Senior Fellow of the Neural Computation program. We thank Matej Usaj (Boone and Andrews Labs, University of Toronto, Canada) for assistance with database management. We also thank the Schuldiner Lab (Weizmann Institute of Sciences, Israel) for providing us with images of their RFP-ORF fusion collection.

## Author contributions

BJA, CB, and BJF designed and supervised the project; YC performed the α-factor imaging screen; BTG generated the WT-2017 strains, performed the imaging, and assisted with the implementation of DeepLoc; JB assisted with early implementations of DeepLoc; OZK implemented DeepLoc; BJA, CB, BJF, BTG, and OZK wrote the manuscript.

## Conflict of interest

The authors declare that they have no conflict of interest.

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
