## [Review Process File · Molecular Systems Biology]

Automated analysis of high-content microscopy data with deep learning

Mr. Oren Kraus, Mr. Ben Gryns, Mr. Jimmy Ba, Yolanda Chong, Brendan Frey, Charles Boone and Brenda J. Andrews

Corresponding author: Charles Boone & Brenda J. Andrews, University of Toronto

Review timeline:	Submission date:	24 January 2017
	Editorial Decision:	02 March 2017
	Revision received:	10 March 2017
	Editorial Decision:	14 March 2017
	Revision received:	16 March 2017
	Accepted:	20 March 2017

Editor: Maria Polychronidou

Transaction Report:

1st Editorial Decision

02 March 2017

Thank you again for submitting your work to Molecular Systems Biology. We have now heard back from the three referees who agreed to evaluate your study. As you will see below, the reviewers appreciate that the proposed approach is going to be of interest for the field. However, they raise a series of concerns, which we would ask you to address in a revision. The reviewers' recommendations are quite clear so I think that there is no need to repeat the points listed below. Of course, please feel free to contact me in case you would like to discuss any of the issues raised by the reviewers.

REFeree REPORTS

Reviewer #1:

In this paper, the authors present a deep learning based approach called DeepLoc for automatically classifying protein localisation in fluorescence images of yeast cells. They test the method do reanalyse an imaging data set they previously produced in their lab (Chong et al. 2015) and to compare it with the previous analysis that was approached using support vector machine (SVM) classifiers (ensLOC). The authors demonstrate that DeepLoc outperforms ensLOC by requiring less training and producing more accurate classifications. They also show that it can be more easily transferred to imaging data sets obtained using different instruments or using cells that differ in their overall morphology (e.g. after treatment with α -factor) or where different fluorophore tag have been used.

This is a well-executed study that demonstrates the utility of deep learning for studies of protein

localisation and quantification. Importantly, it addresses the question of transferability, a typical problem of machine learning based solutions. I also appreciate the fact that the authors made all the data and the code publicly available. Therefore, it will be of interest both for biologist users of machine learning solutions in imaging and for the developers of such approaches.

I only have a few comments regarding the clarity of the descriptions.

When using DeepLoc to study protein localisation upon α -factor treatment the authors write that they "identified 297 proteins (Table S1) whose localization changed significantly". However, in Table S1 I found for 193 out of the 297 proteins that the "predominant untreated localization" and the "predominant alpha factor localization" are the same. Please clarify.

In the following paragraph, the authors write that they used DeepLoc to also extract pixel intensities of alpha treated cells and that these intensities correlated positively with gene expression changes.

The significance of this finding and its connection to the DeepLoc-based localization changes is not clear to me. Were local concentrations measured? If yes, a comparison with gene expression changes does not teach us anything new. If not, how does this profit from the approach presented?

When writing about the transferability of DeepLoc to new microscopy data sets the authors write that they "incorporated five new localization classes (...) (e.g. Cytoplasmic foci, eisosomes, and lipid particles) (...) (Fig. 4A)" that are absent from ensLOC. What exactly are these five new classes? Fig. 4A presumably shows the classes derived by ensLOC since the strains referred to in this section carried RFP-tagged genes. Please clarify.

Finally, as a more general point, I wondered why the authors compared their DeepLoc approach to the SVM based ensLOC rather than other deep learning based approaches for protein localization in yeast such as the ones mentioned in the Introduction. I presume the authors have their reasons but they do not discuss them.

Reviewer #2:

In the submitted manuscript, the authors describe a deep learning approach (DeepLoc) for high-content microscopy. The authors applied DeepLoc on a data set of yeast cells that express GFP-tagged proteins and predict their subcellular localization using a supervised machine learning approach. DeepLoc was benchmarked against a previous machine learning approach the authors developed (ensLoc). A particular application of the DeepLoc networks was the use directly at the pixel level of 'boxed' cells without first segmenting cells and extracting features.

This is a timely manuscript as with the availability of deep learning methods and standardized deep learning software, such as the TensorFlow framework, constructing, training and deploying deep learning algorithms becomes more readily accessible for image analysis applications.

The authors first introduce DeepLoc by benchmarking it against their previously published workflow which used an ensemble of support vector machines (ensLOC) and relies on phenotypic features extracted from segmented cells. They show the advantages to classify cells after training and testing DeepLoc with the data set from the ensLOC publication, both qualitatively and quantitatively. In addition to the comparison to established machine learning approaches and explanations, the second part of the paper focuses on the unique advantages of the methodology. This is done by first, analyzing a data set for which a classification using ensLOC workflow failed and second by analyzing two datasets that have been acquired under changed experimental conditions. For the first part of the analysis, cells with a different morphology than the wild-type cells were classified in the subcellular localization classes. This demonstrated that a classification based on learned parameters of DeepLoc are more robust towards morphological changes of the cells than approaches relying on segmentation. The second analysis demonstrates the adaptability of DeepLoc and the associated workload to achieve this. Transfer learning is applied to adopt the network to data sets that have been acquired using other technical equipment and from other labs. For both analysis workflows, the results using DeepLoc are reported and discussed, direct comparisons to other methods are however not provided. DeepLoc is described as an open-source software that can be easily updated to new tasks and experimental conditions.

Overall, this is a well written manuscript and an interesting adaptation of deep learning networks for high-content screening analysis. The quantitative assessment of the method compared to a previous analysis showed direct evidence for a better performance.

Major points:

- 1) An important advantage of the authors approach compared to previous studies is the avoidance of cell segmentation and feature derivation. However, the reviewer wonders whether this might be restricted to data sets where individual cells can be clearly separated from each other in order to be able to extract "single cell" images. This might not be the case for many image-based high content screens which then again would require cell segmentation.
- 2) In the results the author mention that a smaller training set was used than for training the ensLOC SVM classifier. In the methods part, this is further specified and noted that "positive" examples were sampled for the training set. What does this mean?
- 3) Evaluation of the abundance changes required features derived from segmented cells (CellProfiler features), this disagrees with the general concept of the study to circumvent a cell segmentation.
- 4) It is not clear why the abundance changes were included. It is for example mentioned that proteins rarely change in both, localization and abundance. There should be no striking correlation between protein abundance and localization, accordingly the DeepLoc analysis does not provide any advance for the analysis of abundance changes. Fig 3B: in the main text the authors mention that the micrographs show abundance/localization changes, only localization changes are however mentioned in the figure legend. How does DeepLoc perform without the abundance feature?
- 5) The Figure legend 2 A/B and the corresponding part of the main text are not sufficient to fully understand what is shown.
- 6) The performance of DeepLoc after transfer learning is evaluated with the accuracy only. Why was not precision and recall used as for evaluation of DeepLoc after the initial training?
- 7) The classification of the pheromone screen is not quantitatively evaluated, this should be included in the discussion
- 8) More information and technical details should be provided in the method section about the implementation of DeepLoc.
- 9) Data and software should be made available for download. The Github repository mentioned in the manuscript could not be found. The implementation of the model and the source code was not found by the reviewer and should be provided.

Minor points:

- 1) The manuscript uses quite a bit of technical "jargon" terminology common to the deep learning field but not suitable for a more general audience. The authors should make efforts to explain this better to a broader "systems biology" audience.
- 2) Figure 2A-B could be enhanced by a more quantitative comparison of cluster quality of the two approaches.
- 3) How does the network discover invariant features that allow the identification of spatial compartments? This would be good to explain and discuss in more detail.
- 4) In most cases no exact numbers (e.g. for performance parameters) are given but rather \sim and $<$
- 5) Sheet W1 in table S2 not provided (mentioned in methods: Evaluating DeepLoc Performance)
- 6) In legend of Figure 2A, 256 features are mentioned, however it is difficult to understand how those features are derived as it is not implicitly mentioned in the main text (Information can be found in the methods part and understanding of the deep network approach is required to derive the information)
- 7) In the results part of the pheromone screen "a MAPK" is mentioned, the corresponding figure (Fig. 3B) contains protein names and a yeast biology background is required to figure out which protein the MAPK is
- 8) In the methods part, it is explicitly explained how and to which size (in pixels) the images are cropped. The size of the raw image is however missing.

General remarks:

The conclusions such as the performance in comparison to previous approaches by the authors and the success of the transfer learning are well supported by the data shown in the manuscript.

The manuscript could be in general improved by providing more details about the implementation of the deep learning network and guidance to readers how deep convolutional neural networks can be implemented for other image analysis approaches. It would be also helpful if the authors discuss

their approach in more detail in comparison to other approaches (e.g. Krauss et al., 2016; Duerr et al., 2016).

Reviewer #3:

Although the theory behind deep learning based classification approaches is decades old, driven by increases in computational power, and the assembly of large training data sets, the last few years have witnessed an explosion in the use of deep learning methods in a number of diverse settings. The classification of cellular phenotypes imaged as part of high-throughput microscopy-based screens represents an obvious, but very exciting, application of deep learning tools; as their use could potentially be an ideal means by which to automatically and rapidly categorize phenotypes that can be complex and subtle.

This work by Kraus et al. builds on their previous studies of using deep learning (Kraus et al., Bioinformatics 2016) to classify image-based phenotypes. Here they develop "DeepLoc", a convolution Neural Network (NN) to analyze the distribution of fluorescently-tagged proteins in *Saccharomyces cerevisiae*. The authors make use of a large data set of images of yeast where proteins have been tagged on a genome-scale. In this work the authors primarily compare the performance of DeepLoc to "ensLoc", a previously derived method to classify the distribution of tagged proteins that implements an ensemble of Support Vector Machine (SVM) based classifiers. Furthermore, the authors use DeepLoc to classify protein sub-cellular localization in a new data set where yeast have been treated with alpha-Factor, and on data sets generated by other laboratories.

While I was initially quite excited to see the application of a deep learning tool to an image-based screen, I was underwhelmed by this study. In large part this is due to the fact that, while in some regards DeepLoc shows improved classification performance compared to ensLoc, in many particular cases the performance of DeepLoc is less than impressive. Furthermore, because of the way DeepLoc was implemented, I don't think it represents a truly big step forward in terms of high-content image analysis. Critically, the deployment of DeepLoc did not result in any biological insight to appeal to the broad readership of Molecular Systems Biology. At this time I would recommend publication in a more specialized journal.

Major points:

1) The increased precision of DeepLoc to ensLoc on single cells is particularly impressive "across the board" (ability to classify different sub-cellular localizations). There is also clearly a performance improvement when classifying based on population averages, but it isn't as stunning as when analyzing single cells, and is based largely on DeepLoc's ability to better classify 4-5 phenotypes.

In my mind this is really the most impressive result of the work. But I don't think it represents a truly significant, and novel, impact on the field of image analysis and/or functional genomics.

I do wonder how much of the improved precision is due to the differences in segmentation. Could it be that ensLOC struggles to classify certain phenotypes using single cell analysis because of segmentation issues and/or morphological differences between cells that are somewhat reduced when using bounding boxes - which ignore cell morphology?

In fact, I would expect that differences between a bounding box and cell segmentation approach might be washed away when looking at population averages. Can the authors account for these differences to show that it indeed is the classifier, and not simply the segmentation that is driving performance improvement - especially on single cells?

I would really like to see the performance of DeepLoc on whole images (no bounding box), because I think the ability to classify phenotypes in the absence of segmentation is what the field is really looking for.

2) The authors argue that analysis of the alpha-factor screens is a powerful application of DeepLoc

because there is no need "for additional, non-wild-type training, while re-implementing a SVM ensemble would have necessitated weeks of training and optimization."

To me the analysis of this screen (and the re-analysis of other screens below) doesn't really demonstrate the broad utility of this approach. In the case of the alpha-factor screen, I wouldn't expect DeepLoc's classification method to be particularly "challenged" because the cells are not segmented in a way that would confound the analysis. In fact, I might even predict that an ensemble-based method would perform equally well between untreated and alpha-factor treated conditions if a bounding box type segmentation was used.

3) The authors then test DeepLoc on additional data sets, but I think here the results are far less impressive. Accuracies of ~40% (or even less sometimes) are hardly evidence of significant methodological improvements. To put it another way, I still think any biologist who wants to perform a rigorous analysis of these data, or any other new data sets would be better off spending time developing an ensemble based method than using DeepLoc.

4) Finally, there is no real biological insight gained from the application of DeepLoc. So while the method may provide a faster means by which to analyze data, in the absence of such insight its not clear to me why it should be used.

5) The current manuscript seems to largely ignore the extensive amount of work that has been done on quantifying sub-cellular localization of proteins using other methods. In fact, comparing DeepLoc to some of these other tools (other than just ensLoc) may be warranted.

6) Why does this work represent an significant advance over the authors' recently published work in Bioinformatics?

Additional specific comments:

- Why did the authors test their method on only a subset of the data set used for training ensLoc (~22,000 out of 70,000 images)? It is acceptable to train the CNN on a balanced subset, but why was the entire set not used to measure performance? Such as test would provide a better estimation of the generalizability of the trained features.

- Related to the above point, the details on choosing the subset are not provided.

- The authors generate patches of 64x64 pixels that are centered on single cells. More information should be provided on the size of the cell and how often the selected patch size does not cover the entire cell segment.

- It will be useful if an analysis of when CNN fails is provided.

- When using DeepLoc in classifying cells in response to alpha factor the authors state "DeepLoc produced reasonable protein classification for single cells within hours ...". Exact numbers on the average precision of applying DeepLoc should be provided given a representative sample.

1st Revision - authors' response

10 March 2017

Reviewer #1

Comments to the Authors:

When using DeepLoc to study protein localisation upon α -factor treatment the authors write that they "identified 297 proteins (Table S1) whose localization changed significantly". However, in Table S1 I found for 193 out of the 297 proteins that the "predominant untreated localization" and the "predominant alpha factor localization" are the same. Please clarify.

We thank the reviewer for this comment, and we apologize for any confusion. This table (now Table EV1) lists 297 proteins in which a significant localization change occurred after alpha-factor treatment. In this table we have provided quantitative t-test scores for each subcellular localization class for each of the 297 cell populations. We have also listed the predominant localizations in untreated as well as alpha-factor treated conditions, which are simply the classes with the highest localization scores. In 193 of these cell populations, the predominant localization remains the same across conditions, though the scores were significantly different.

For example, cells expressing *YOXI-GFP* in alpha-factor became significantly more cytoplasmic (t-test “Cytoplasm” = -17.4) and significantly less nuclear (t-test “Nuclei” = 10.9). However, in both untreated and alpha-factor conditions, the predominant localization was still “Cytoplasm”. All scores are available at “<http://cyclops.cabr.utoronto.ca>” and can be validated by searching “Yox1”.

To clarify, we have added the following to *Identifying Significant Localization and Abundance Changes in α -Factor* in the Materials and Methods section (Page 17, Paragraph 1):

“For some of these proteins, the dominant localization was the same in both conditions although the distributions of localization scores differed significantly.”

In the following paragraph, the authors write that they used DeepLoc to also extract pixel intensities of alpha treated cells and that these intensities correlated positively with gene expression changes. The significance of this finding and its connection to the DeepLoc-based localization changes is not clear to me. Were local concentrations measured? If yes, a comparison with gene expression changes does not teach us anything new. If not, how does this profit from the approach presented?

We agree with the reviewer that the incorporation of abundance measurements after alpha-factor treatment may seem out of place in our study, which primarily emphasizes protein localization data. There are number of reasons why we included protein abundance measurements. Firstly, the alpha-factor dataset has not been previously published; here, we are providing a quantitative repository of changes in both localization and abundance for the yeast community to further investigate and to serve as a benchmark for future research. We also provided all of our quantitative measurements on the Cyclops database (<http://cyclops.cabr.utoronto.ca>) where similar high-content screening data is published on both protein localization and abundance. The inclusion of protein abundance information therefore makes our assessment complementary to the other screening analyses in our database. Finally, because this screen is previously unpublished, and we were unable to obtain high-throughput, quantitative protein abundance or localization changes in alpha-factor, we felt our best option for validating the efficacy of our screen was to benchmark against gene expression. In this instance, we found that many of the proteins that increase in their abundance after alpha-factor treatment are also known to be regulated at the level of transcription.

To help clarify our intentions, we have included the following in the *Using DeepLoc to Identify Protein Dynamics in Response to Mating Pheromone* section of the Results (Page 8, Paragraph 3):

“While unrelated to the localization analysis by DeepLoc, this evaluation of protein abundance further validates the effectiveness of our screening protocol; it also provides a complementary overview of proteomic responses to those made by Chong et al. (2015) on the Cyclops database.”

When writing about the transferability of DeepLoc to new microscopy data sets the authors write that they “incorporated five new localization classes (...) (e.g. Cytoplasmic foci, eisosomes, and lipid particles) (...) (Fig. 4A)” that are absent from ensLOC. What exactly are these five new classes? Fig. 4A presumably shows the classes derived by ensLOC since the strains referred to in this section carried RFP-tagged genes. Please clarify.

To clarify, this dataset (WT-2017) is not the same dataset that was previously analyzed using ensLOC, but rather a new dataset that we generated in our lab. As mentioned in the text, this new dataset was also generated in untreated conditions, but on a different microscope and with different red fluorescent markers (e.g. an mCherry-tagged histone protein to mark the nucleus of the cell).

When analyzing this dataset, we wanted to be even more comprehensive in our coverage of subcellular localization classes. As mentioned in the text, we also wanted to see if DeepLoc would be able to distinguish classes that look highly similar under manual inspection, but are known to have different biological roles. For these reasons, we included five classes in this analysis that had not been incorporated into ensLOC. These classes are: Cytoplasmic Foci, Eisosomes, Lipid Particles, Bud Site, and Punctate Nuclear.

Finally, as a more general point, I wondered why the authors compared their DeepLoc approach to the SVM based ensLOC rather than other deep learning based approaches for protein localization in yeast such as the ones mentioned in the Introduction. I presume the authors have their reasons but they do not discuss them.

We thank the reviewer for this comment. There are a number of reasons why we chose to compare DeepLoc performance with the SVM-based ensLOC. Firstly, with respect to traditional machine learning approaches (i.e. not employing deep learning), Koh *et al.* (2015) compared ensLOC to previous methods for quantitatively analyzing protein localization in yeast and reported that ensLOC outperforms the previous approaches. Here, we treat ensLOC as the current gold-standard for yeast images as it is the only quantitative method to be developed and reliably deployed to several other proteome wide perturbation screens.

With respect to other deep learning-based approaches, there were two that we mentioned in the Introduction:

1. The Kraus *et al.* (2016) paper used similar yeast images, but in this instance models were trained on whole images and does not allow for *single cell* comparisons. This approach does not provide predictions for *single cells* and cannot be directly compared to the ensLOC performance on the same individually labeled cells.
2. Parnamaa and Parts (2016) also used a similar dataset in their pre-print, but again, this set was not labeled at the single cell level.

We clarify these points regarding other deep learning analyse in the Discussion (Page 11, Paragraph 3):

“These results differentiate DeepLoc from previous implementations of deep learning for high-throughput cell image data. Recent publications demonstrate the improved accuracy achieved by deep learning based classifiers for high content screening (Kraus *et al.*, 2016; Parnamaa & Parts, 2016; Dürr & Sick, 2016) and for imaging flow cytometry (Eulenberg *et al.*, 2016). These reports validate their proposed models on held out test sets from the same source as the training data and typically evaluate less phenotypes than DeepLoc (i.e. 4 mechanism of action clusters in Dürr & Sick (2016) and 5 cell cycle stages in Eulenberg *et al.* (2016)). In Kraus *et al.* (2016), we describe a deep learning framework for classifying whole microscopy images that is not designed to classify single cells. Here we train DeepLoc on 15 sub-cellular localizations classes from one genome-wide screen, deploy DeepLoc to a second genome-wide screen of cells with substantially altered cell morphology that was not amenable to classification with EnsLoc, and then use transfer learning to deploy DeepLoc to image sets that were screened differently than the training set with minimal additional labeling.”

Reviewer #2

Major Points:

1. An important advantage of the authors approach compared to previous studies is the avoidance of cell segmentation and feature derivation. However, the reviewer wonders whether this might be restricted to data sets where individual cells can be clearly separated from each other in order to be

able to extract "single cell" images. This might not be the case for many image-based high content screens which then again would require cell segmentation.

We thank the reviewer for this comment. To clarify, in many instances the bounding box that is centered on a single cell will still contain additional cells in the field, and we see that these images still work well in our framework. Below, we provide an image of cells used as training input for the "Cell Periphery" class in DeepLoc; you will observe that many of the images used for training contain multiple cells or parts of other cells. We feel that this provides some evidence that DeepLoc will still be able to carry out classification tasks containing multiple cells or cells that cannot be segmented.

To clarify, we have included the following in the *Training and Validating a Deep Neural Network (DeepLoc) for Classifying Protein Subcellular Localization in Budding Yeast* section of the Results (Page 5, Paragraph 1):

“However, instead of training a classifier on feature sets extracted from segmented cells, we trained DeepLoc directly on a defined region of the original microscopy image centered on a single cell, but often containing whole, or partial cells in the periphery of the box. The use of these “bounding boxes” removes the sensitivity of the image analysis to the accuracy of segmentation that is typical of other machine learning classifiers.”

2. In the results the author mention that a smaller training set was used than for training the ensLOC SVM classifier. In the methods part, this is further specified and noted that "positive" examples were sampled for the training set. What does this mean?

To clarify, the ensLOC classifier consisted of training 60 binary SVMs each with a unique training set of positive and negative samples (e.g. for the “Cytoplasm” classifier, there were images of single segmented cells with GFP-fusion proteins that localize in the cytoplasm (positive samples)) and then single segmented cells with GFP-fusions that localize to all other classes were labeled as negative samples. In contrast, we trained DeepLoc as one model in a multi-class setting. This meant that we could not reliably use the ‘negative’ samples from the original training set, because they weren’t annotated to belong to a particular localization class (rather they were annotated as not belonging to particular class and were a mix of many different localizations). We further reduced the dataset size by subsampling classes that had many samples.

To clarify, we have included the following in the *Training DeepLoc* subsection in the Materials and Methods section (Page 15, Paragraph 2):

“The original labeled dataset was composed of 60 sub-datasets, each containing ‘positive’ and ‘negative’ samples, to train the 60 binary SVM classifiers used in ensLOC. Instead of using all of the ~70,000 cells previously annotated, we sampled only positive examples such that each localization compartment contained ~500-1,500 cells and we trained DeepLoc as a multi-class

classifier.”

3. Evaluation of the abundance changes required features derived from segmented cells (CellProfiler features), this disagrees with the general concept of the study to circumvent a cell segmentation.

We agree with the reviewer’s comment. As mentioned in our response to Reviewer 1, there are a variety of reasons why we incorporated the abundance data into our analysis of the alpha-factor screen (benchmarking the dataset, comparability with the other analyses on the Cyclops database, completeness for the yeast community). This analysis was separate from DeepLoc, which uses neither the abundance data, nor the other extracted features in its classifications.

To help clarify, we have included the following in the *Using DeepLoc to Identify Protein Dynamics in Response to Mating Pheromone* section of the Results (Page 8, Paragraph 3):

“While unrelated to the localization analysis by DeepLoc, this evaluation of protein abundance further validates the effectiveness of our screening protocol; it also provides a complementary overview of proteomic responses to those made by Chong et al. (2015) on the Cyclops database.”

4. It is not clear why the abundance changes were included. It is for example mentioned that proteins rarely change in both, localization and abundance. There should be no striking correlation between protein abundance and localization, accordingly the DeepLoc analysis does not provide any advance for the analysis of abundance changes. Fig 3B: in the main text the authors mention that the micrographs show abundance/localization changes, only localization changes are however mentioned in the figure legend. How does DeepLoc perform without the abundance feature?

Please see above.

5. The Figure legend 2 A/B and the corresponding part of the main text are not sufficient to fully understand what is shown.

We appreciate this feedback from the reviewer. While the technical details are thoroughly explained in the Materials and Methods section, we have amended the main text in an effort to simplify our analysis.

The following has been amended/added to the *Visualizing Network Features* section of the Results (Page 6, Paragraph 1):

“To address whether this difference was relevant in our experiments, we visualized the activations of the final convolutional layer in 2D using t-distributed stochastic neighbor embedding (t-SNE) (Maaten & Hinton, 2008) for a single cell test set (Fig. 2A). t-SNE is a popular non-linear dimensionality reduction algorithm often used to visualize the structure within high dimensional data in 2D or 3D space.”

6. The performance of DeepLoc after transfer learning is evaluated with the accuracy only. Why was not precision and recall used as for evaluation of DeepLoc after the initial training?

To clarify, we used precision/recall in the initial evaluation of DeepLoc because some classes were heavily imbalanced. For example, the spindle class only had 185 single cell training samples while other classes had a maximum of 1,500 samples.

In contrast, when analyzing transfer learning we controlled the number of samples per class and ensured that the classes were balanced during training. Therefore it was more appropriate to report accuracy for this evaluation. For the sampling sizes for which we show the confusion matrices, one can see the performance of different classes. We included the prediction accuracy calculations along the y-axis of these plots.

Furthermore, because many proteins localize to multiple classes, we felt that our approach made the most sense.

7. The classification of the pheromone screen is not quantitatively evaluated, this should be included in the discussion.

We appreciate this feedback from the reviewer; unfortunately, a quantitative evaluation of the pheromone screen would have necessitated extensive additional analyses as we did not have a manually annotated set of single cell images, nor a systematic gold-standard for protein localization after alpha-factor treatment. Note that in our comparison to ensLOC, we had access to the single cell training sets that were previously generated by our group, but which took a substantial amount of time to manually annotate. We also had gold-standard protein localization data from previous manual assessments of the ORF-GFP Fusion Collection in untreated conditions.

Instead we aimed to show that the model could be deployed to a new set that was not previously labeled. We manually confirmed the results from DeepLoc and all the images and predictions are reported in the Cyclops database (<http://cyclops.cabr.utoronto.ca>). In addition, we performed enrichment analysis on the 100 proteins representing the most substantial localization changes and found that many of these are already implicated in the mating response program. We hope that our analysis of the proteome in alpha-factor treated cells will catalyse further experiments and validation by the yeast community.

8. More information and technical details should be provided in the method section about the implementation of DeepLoc.

We thank the reviewer for this comment. We added an expanded view figure (Figure EV1) to better illustrate how computations are carried out in the convolutional neural network.

We believe that the details provided in Materials and Methods/*Training DeepLoc* section are sufficient to reproduce the DeepLoc model. We also provide the all the datasets and source code used to implement DeepLoc. We apologize for not including the temporary link we prepared for the code in the original submission (it is currently hosted at: http://spidey.cabr.utoronto.ca/~okraus/DeepLoc_Supplemental_Software.zip). After publication we will host it on github as well so that we can continue to update the repository and track issues. Readers interested in more technical details than provided in the Materials and Methods section can check the repository and run the model on their own machines.

9. Data and software should be made available for download. The Github repository mentioned in the manuscript could not be found. The implementation of the model and the source code was not found by the reviewer and should be provided.

We sincerely apologize for this oversight. We intend to share the code in github post-publication and forgot to change the link in the paper to the temporary location of the source code. The code is currently hosted here:

http://spidey.cabr.utoronto.ca/~okraus/DeepLoc_Supplemental_Software.zip

Minor Points:

1. The manuscript uses quite a bit of technical "jargon" terminology common to the deep learning field but not suitable for a more general audience. The authors should make efforts to explain this better to a broader "systems biology" audience.

We appreciate this feedback from the reviewer. In an effort to clarify the technical jargon in the main text we have added an expanded view figure (Figure EV1) to better illustrate how computations are carried out in the convolutional neural network; in this figure we did our best to

explain these complex concepts in more general terms. We have also referred the reader to multiple review papers on machine learning and neural networks in the introduction. We hope that these changes will be sufficient for the systems biology audience.

2. Figure 2A-B could be enhanced by a more quantitative comparison of cluster quality of the two approaches.

To address this concern, we calculated the Davies-Bouldin Index for the two feature representations. The index is typically used to compare different clustering algorithms. According to the metric, clustering results with low intra-cluster and high inter-cluster distances produce lower index values and are preferred. Here we compare the feature representations in figures 2A/B and treat the real localization labels as the cluster assignment for each data point. The neural network representation we use is network activations taken from the 8th layer (last convolutional layer before the 2 fully connected layers). For CellProfiler we use 313 extracted features.

The Index scores we calculated are:
DeepLoc: 2.33, CellProfiler: 4.36

However, an explanation of this metric requires additional “jargon-heavy” explanation in the main text of the manuscript. We feel that the figure and corresponding legend sufficiently illustrate our point to the reader.

3. How does the network discover invariant features that allow the identification of spatial compartments? This would be good to explain and discuss in more detail.

We thank the reviewer for this feedback. As mentioned above, we have added an expanded view figure (Figure EV1) to better illustrate how computations are carried out in the convolutional neural network, including details regarding the discovery of invariant features.

4. In most cases no exact numbers (e.g. for performance parameters) are given but rather \sim and $<$.

We have updated numerous values throughout the results section of the manuscript to reflect exact values.

5. Sheet W1 in table S2 not provided (mentioned in methods: Evaluating DeepLoc Performance)

We apologize for any confusion. This sheet is in the supplement for the Chong *et al.* (2015) paper. We are not referring to our own supplementary material, though this will likely be more clear now that our own supplement will be labeled as an “Expanded View” with “EV” rather than “S”.

To clarify further, we have added the following to the *Evaluating DeepLoc Performance* section of the Materials and Methods (Page 16, Paragraph 2):

“To compare with Chong *et al.* (2015), we used the values reported in the WT1 sheet of Table S2 in their publication.”

6. In legend of Figure 2A, 256 features are mentioned, however it is difficult to understand how those features are derived as it is not implicitly mentioned in the main text (Information can be found in the methods part and understanding of the deep network approach is required to derive the information)

We thank the reviewer for this feedback. As mentioned above, we have added an expanded view figure (Figure EV1) to better illustrate how computations are carried out in the convolutional neural network. In the *Training and Validating a Deep Neural Network (DeepLoc) for Classifying Protein Subcellular Localization in Budding Yeast* section of the Results (Page 5, Paragraph 1), we direct readers to this figure and its corresponding description:

“We provide a brief overview of convolutional neural networks in Figure EV1 and refer readers to Goodfellow *et al.* (2016) and LeCun *et al.* (2015) for a more thorough introduction.”

7. In the results part of the pheromone screen "a MAPK" is mentioned, the corresponding figure (Fig. 3B) contains protein names and a yeast biology background is required to figure out which protein the MAPK is.

We apologize for any confusion, however, we clearly stated in the main text that the MAPK is the yeast protein Kss1. We wrote: “Importantly, DeepLoc identified novel movements of proteins already implicated in the mating response, such as the movement of Kss1, a MAPK that functions primarily to regulate filamentous growth, from the nucleus to the cytoplasm.” Based on this description, the reader can locate “Kss1” in the figure and know that this is the MAPK described in the main text.

8. In the methods part, it is explicitly explained how and to which size (in pixels) the images are cropped. The size of the raw image is however missing.

We apologize for this oversight. We have added this information into the *Live Cell Image Acquisition* section of the Material and Methods (Page 14, Paragraph 2-3):

“A total of 4 images were acquired in each channel (1349x1004 pixels), resulting in a total screening time of ~40 minutes per 384-well plate.”

“A total of 10 images (1338x1003 pixels) were acquired in each channel, resulting in a total screening time of ~100 minutes per 384-well plate.”

General Remarks:

The manuscript could be in general improved by providing more details about the implementation of the deep learning network and guidance to readers how deep convolutional neural networks can be implemented for other image analysis approaches.

It would be also helpful if the authors discuss their approach in more detail in comparison to other approaches (e.g. Kraus *et al.*, 2016; Duerr *et al.*, 2016).

We appreciate this feedback. We have added the following to the Discussion (Page 11, Paragraph 3):

“These results differentiate DeepLoc from previous implementations of deep learning for high-throughput cell image data. Recent publications demonstrate the improved accuracy achieved by deep learning based classifiers for high content screening (Kraus *et al.*, 2016; Pärnamaa & Parts, 2016; Dürr & Sick, 2016) and for imaging flow cytometry (Eulenberg *et al.*, 2016). These reports validate their proposed models on held out test sets from the same source as the training data and typically evaluate less phenotypes than DeepLoc (i.e. 4 mechanism of action clusters in Dürr & Sick (2016) and 5 cell cycle stages in Eulenberg *et al.* (2016)). In Kraus *et al.* (2016), we describe a deep learning framework for classifying whole microscopy images that is not designed to classify single cells. Here we train DeepLoc on 15 sub-cellular localizations classes from one genome-wide screen,

deploy DeepLoc to a second genome-wide screen of cells with substantially altered cell morphology that was not amenable to classification with EnsLoc, and then use transfer learning to deploy DeepLoc to image sets that were screened differently than the training set with minimal additional labeling.”

Reviewer #3

Major Points:

1. The increased precision of DeepLoc to ensLoc on single cells is particularly impressive "across the board" (ability to classify different sub-cellular localizations). There is also clearly a performance improvement when classifying based on population averages, but it isn't as stunning as when analyzing single cells, and is based largely on DeepLoc's ability to better classify 4-5 phenotypes.

In my mind this is really the most impressive result of the work. But I don't think it represents a truly significant, and novel, impact on the field of image analysis and/or functional genomics.

We hope that the revisions we have made to our paper in response to all reviewers' comments will help highlight the importance of our work for the image analysis and functional genomics community. We find that the classification performance at the single cell level is improved for every localization class and for most of them by a substantial margin. The improvements in protein level assignments may be less pronounced because these annotations are aggregated over cell populations while Chong *et al.* (2015) used an additional training step to calculate optimal thresholds for assigning sub-cellular localizations to proteins; in our analysis we simply calculated the mean prediction for each class across the cellular population; [2] the significant improvements for the 4-5 localizations classes mentioned are important for studying yeast protein dynamics, as failing to classify these classes correctly can result in substantially overlooking dynamics related to entire bio-processes or subcellular localizations.

I do wonder how much of the improved precision is due to the differences in segmentation. Could it be that ensLOC struggles to classify certain phenotypes using single cell analysis because of segmentation issues and/or morphological differences between cells that are somewhat reduced when using bounding boxes - which ignore cell morphology?

We appreciate this comment from the reviewer; however, in the Chong *et al.* (2015) publication, they mention that ensLOC included quality control classifiers to filter out mis-segmented, ghost objects, and dead cells. In the supplement for their work, they mention that 3-10% of objects were filtered out by these classifiers, and that they required at least 15 cells per condition for their analyses. Taking these figures into account, sub-cellular localization classes that were included in ensLOC were likely not hindered by segmentation errors.

Interestingly, the reviewer's point is a strength of DeepLoc, as we do not need to extract features that are dependent on segmentation performance, making DeepLoc robust to segmentation errors.

In fact, I would expect that differences between a bounding box and cell segmentation approach might be washed away when looking at population averages. Can the authors account for these differences to show that it indeed is the classifier, and not simply the segmentation that is driving performance improvement - especially on single cells?

While it is difficult to extract the exact same features from bounding boxes, we did evaluate the performance of a fully connected neural network with two hidden layers (same architecture as the fully connected layers in DeepLoc) on features extracted from CellProfiler. Here we see that the

performance is drastically improved for single cells over the ensemble of 60 binary SVMs used in ensLOC. We still see an overall improvement by training a convolutional network directly on the pixel intensity data (DeepLoc). These results show that the neural network classifier is a powerful alternative to SVMs and that training convolutional networks directly on pixel intensity data performs best. We feel that this additional analysis is unnecessary for our publication, but we can include it in the supplement if the editors feel it is essential.

I would really like to see the performance of DeepLoc on whole images (no bounding box), because I think the ability to classify phenotypes in the absence of segmentation is what the field is really looking for.

Our previous publication, Kraus *et al.* (2016), describes a convolutional architecture specifically for classifying whole images without segmentation. Although very useful, this model wasn't designed to output classifications for *single cells*, which is a requirement for many high content-screening experiments in order to assess phenotype heterogeneity.

2. The authors argue that analysis of the alpha-factor screens is a powerful application of DeepLoc because there is no need "for additional, non-wild-type training, while re-implementing a SVM ensemble would have necessitated weeks of training and optimization."

To me the analysis of this screen (and the re-analysis of other screens below) doesn't really demonstrate the broad utility of this approach. In the case of the alpha-factor screen, I wouldn't expect DeepLoc's classification method to be particularly "challenged" because the cells are not segmented in a way that would confound the analysis. In fact, I might even predict that an ensemble-based method would perform equally well between untreated and alpha-factor treated conditions if a bounding box type segmentation was used.

Our previous pipeline (ensLOC) failed to classify the alpha-factor screen and this screen was therefore left out of the Chong *et al.* (2015) publication and the Cyclops database (until it was recently analyzed with DeepLoc).

Using a different segmentation approach (i.e. using bounding boxes) would change many of the values of the extracted features and require retraining ensLOC once again. Given that we have shown the improved performance of neural networks on extracted features compared to the SVM ensemble, we still believe that, regardless of the segmentation approach, DeepLoc will outperform

any existing classification approach based on extracting hand crafted features. Further, even if the performances were somewhat comparable, DeepLoc is a much more efficient model to train and deploy. DeepLoc is a single multiclass classifier requiring much less training data than the ensemble of 60 binary SVMs trained on 60 unique datasets with over 70,000 cells overall.

3. The authors then test DeepLoc on additional data sets, but I think here the results are far less impressive. Accuracies of ~40% (or even less sometimes) are hardly evidence of significant methodological improvements. To put it another way, I still think any biologist who wants to perform a rigorous analysis of these data, or any other new data sets would be better off spending time developing an ensemble based method than using DeepLoc.

Here our goal was to demonstrate how quickly DeepLoc can be transferred to a new dataset. In Figure 4D we show the confusion matrix after updating DeepLoc with only 5 samples per localization class. Although some classes perform around 40% accuracy half of the classes perform at an accuracy of 70% or greater. We doubt another method could reach the same overall performance given only 5 samples per class. This feature of DeepLoc is largely due to the fact that the network learned to represent many patterns that are relevant to protein sub-cellular localization from the Chong *et al.* dataset. For practical deployment of DeepLoc to new screens, we still recommend training with more single cell samples per class (~100). This dataset size is still much smaller than that used by Chong *et al.* or DeepLoc without transfer learning.

4. Finally, there is no real biological insight gained from the application of DeepLoc. So while the method may provide a faster means by which to analyze data, in the absence of such insight its not clear to me why it should be used.

As our paper describes a new computational tool, our main focus was not on the new biological insights that can be mined from our data. We note that we not only use DeepLoc to analyse published datasets (from our lab and the Schuldiner lab), but also analyse two unpublished datasets, which are now available to the yeast community for further analyses. In particular, we have used DeepLoc to assess a mating pheromone screen that could not be analyzed and included in our previous publication in *Cell* (2015). This previous publication is an important resource for the systems biology community as it was the first quantitative assessment of protein localization and abundance on a proteome-wide scale. The method we described here overcomes many of the computational barriers faced in quantitatively analyzing proteome wide screens, including overcoming sensitivities to segmentation and feature extraction pipelines, and providing a much simpler and more accurate classifier than large ensembles of SVMs. The screen we analyzed with DeepLoc (that could not be analyzed with ensLOC) provides valuable insight into the yeast mating process, including the identification of 300 proteins for which sub-cellular localization changes significantly in response to alpha-factor, some which are previously uncharacterized proteins. We provide all the localization change predictions and abundance data from this screen as a resource to the yeast community in the Cyclops database. We hope that other labs conducting protein localization screens in yeast will adopt DeepLoc and update it for their own experiments.

5. The current manuscript seems to largely ignore the extensive amount of work that has been done on quantifying sub-cellular localization of proteins using other methods. In fact, comparing DeepLoc to some of these other tools (other than just ensLoc) may be warranted.

We appreciate this feedback from the reviewer. As mentioned in previous responses above, we treat ensLOC as the gold-standard in quantifying protein localization since Koh *et al.* (2015) demonstrated its enhanced performance over earlier methods.

6. Why does this work represent a significant advance over the authors' recently published work in Bioinformatics?

The Bioinformatics, 2016 paper describes a neural network architecture for classifying whole microscopy images with whole image level annotations. Although this architecture is powerful for high-content screening analysis, it was not designed to provide single cell predictions. Here we

describe DeepLoc, a deep convolutional network for classifying protein localization in images of single cropped cells. We thoroughly compared DeepLoc to ensLOC on the tasks of single cell classification and protein level annotations. We subsequently show DeepLoc's ability to classify divergent image sets without the need for substantial tuning and training, making it an invaluable tool for the imaging community to rapidly analyze their datasets.

To clarify, we added the following text to the Discussion section (Page 11, Paragraph 3):

“These results differentiate DeepLoc from previous implementations of deep learning for high-throughput cell image data. Recent publications demonstrate the improved accuracy achieved by deep learning based classifiers for high content screening (Kraus *et al.*, 2016; Pärnamaa & Parts, 2016; Dürr & Sick, 2016) and for imaging flow cytometry (Eulenberg *et al.*, 2016). These reports validate their proposed models on held out test sets from the same source as the training data and typically evaluate less phenotypes than DeepLoc (i.e. 4 mechanism of action clusters in Dürr & Sick (2016) and 5 cell cycle stages in Eulenberg *et al.* (2016)). In Kraus *et al.* (2016), we describe a deep learning framework for classifying whole microscopy images that is not designed to classify single cells. Here we train DeepLoc on 15 sub-cellular localizations classes from one genome-wide screen, deploy DeepLoc to a second genome-wide screen of cells with substantially altered cell morphology that was not amenable to classification with EnsLoc, and then use transfer learning to deploy DeepLoc to image sets that were screened differently than the training set with minimal additional labeling.”

Additional Specific Comments:

Why did the authors test their method on only a subset of the data set used for training ensLoc (~22,000 out of 70,000 images)? It is acceptable to train the CNN on a balanced subset, but why was the entire set not used to measure performance? Such as test would provide a better estimation of the generalizability of the trained features.

We appreciate this feedback from the reviewer. As we mentioned in responses to the other reviewers, ensLOC was trained as an ensemble of 60 binary SVMs, requiring 60 unique datasets with positive and negative samples for each classifier; since we trained DeepLoc in a multi-class setting, we could only use the positive samples from the original dataset (as the negative samples are simply labeled as not belonging to a localization class).

Related to the above point, the details on choosing the subset are not provided.

After selecting the usable data from the 70,000 manually labeled images, we realized that some classes had many more labeled samples. We limited the maximum number of sample cells per class to 1,500.

The authors generate patches of 64x64 pixels that are centered on single cells. More information should be provided on the size of the cell and how often the selected patch size does not cover the entire cell segment.

Yeast cells change in their size over cell cycle progression but average ~49 pixels along the major axis, and 37 pixels along the minor axis of the cell. We chose to use 64x64 pixels based on these measurements and after substantial trial and error during image analysis. As the network is trained on a variety of cells and orientations, occasional cropping of a segment of a cell, or the inclusion of neighboring cells doesn't significantly affect the training performance. See the image below for training samples from the Cell Periphery class. Although the network sees a variety of sizes and orientations, it learns that a ring pattern in the green channel is common these images and learns to recognize the pattern as a discriminative feature.

To address the reviewer's concern we have added the following into the *Training DeepLoc* section of the Materials and Methods (Page 16, Paragraph 2):

“Yeast cells change in their size over cell cycle progression but average ~49 pixels along the major axis, and 37 pixels along the minor axis of the cell.”

It will be useful if an analysis of when CNN fails is provided.

This information is provided in the error matrices in Figures 4D/E and 5D/E for localization classes transfer learning performs more poorly (i.e. incorrect assignment of a protein to another class).

When using DeepLoc in classifying cells in response to alpha factor the authors state "DeepLoc produced reasonable protein classification for single cells within hours ...". Exact numbers on the average precision of applying DeepLoc should be provided given a representative sample.

As mentioned in previous responses above, we did not have a manually labeled set for the pheromone screen. Instead we aimed to show that the model could be deployed to a new set that was not previously labeled. We manually confirmed the results from DeepLoc and all the images and predictions are reported in Cyclops (<http://cyclops.ccb.utoronto.ca>). In addition, we performed enrichment analysis on the 100 proteins representing the most substantial localization changes and found that many of these are already implicated in the mating response program. As we included quantitative evaluations based on manually labeled single cells for the Chong *et al.* data, as well as the two new datasets we used for transfer learning, we feel that quantifying the performance of DeepLoc on the alpha-factor screen with additional labeling would not add much value to the analysis.

2nd Editorial Decision

14 March 2017

Thank you for sending us your revised manuscript. We are satisfied with the modifications made and we think that the study is now suitable for publication.

Before we formally accept the manuscript, we would like to ask you to address the following editorial issue in a minor revision:

- In the Data Availability Section: Please include links providing direct access to the newly generated datasets (α -factor screen) in the Cyclops database and to the software at GitHub. In order to ensure long-term archival alongside the paper, we would also ask you to provide the DeepLoc code as a .zip file labeled Computer Code EV1.

2nd Revision - authors' response

16 March 2017

We have amended our manuscript and supporting material to meet the specifications you outlined in your previous email, with one exception: Unfortunately, because of the way that the Cyclops database is set up, we are unable to provide a direct link to the location of the alpha-factor images and data. Instead, we have included the following description in the "Data Availability" section to guide the reader in navigating Cyclops:

"The results and images for the α -factor screen are available on the Cyclops Database. Here, individual proteins can be queried using the search function, after which corresponding localization and abundance data from our analysis can be accessed under the "DeepLoc" subheading. Under this subheading, the data from our three untreated conditions (WT1, WT2, WT3) as well as the three α -factor time-points (AF100, AF140, AF180) is available for both localization and abundance. In addition, individual micrographs can be accessed under the "Retrieve micrographs from other screen" tab, by selecting "AF100", "AF140", or "AF180": (<http://cyclops.cabr.utoronto.ca>). Raw images will be made available upon request."

3rd Editorial Decision

20 March 2017

Thank you again for sending us your revised manuscript. We are now satisfied with the modifications made and I am pleased to inform you that your paper has been accepted for publication.

Corresponding Author Name: Brenda J. Andrews

Manuscript Number: MSB-17-7551